# The Implications of Local Correlation on Learning Some Deep Functions

**Eran Malach**
School of Computer Science,
The Hebrew University, Israel.
eran.malach@mail.huji.ac.il

**Shai Shalev-Shwartz**
School of Computer Science,
The Hebrew University, Israel.
shais@cs.huji.ac.il

## Abstract

It is known that learning deep neural-networks is computationally hard in the worst-case. In fact, the proofs of such hardness results show that even weakly learning deep networks is hard. In other words, no efficient algorithm can find a predictor that is slightly better than a random guess. However, we observe that on natural distributions of images, small patches of the input image are correlated to the target label, which implies that on such natural data, efficient weak learning is trivial. While in the distribution-free setting, the celebrated boosting results show that weak learning implies strong learning, in the distribution-specific setting this is not necessarily the case. We introduce a property of distributions, denoted "local correlation", which requires that small patches of the input image and of intermediate layers of the target function are correlated to the target label. We empirically demonstrate that this property holds for the CIFAR and ImageNet data sets. The main technical results of the paper is proving that, for some classes of deep functions, weak learning implies efficient strong learning under the "local correlation" assumption.

## 1 Introduction and Motivation

It is well known (e.g. [26]) that while deep neural-networks can **express** any function that can be run efficiently on a computer, in the general case, **learning** neural-networks is computationally hard. Despite this theoretic pessimism, in practice, deep neural networks are successfully trained on real world datasets. Bridging this theoretical-practical gap seems to be the holy grail of theoretical machine learning nowadays. Maybe the most natural direction to bridge this gap is to find a **property** of data distributions that determines whether learning them is computationally easy or hard. The goal of this paper is to propose such a property.

To start off, let us take a closer look into computational hardness results on learning neural-networks. Over the years, the theoretical machine learning community has established many such hardness results, drawing from different hardness assumptions [26, 29, 9, 33]. While these results differ in their technical details, they all share one thing in common: they all show that **weakly** learning neural-networks is computationally hard. That is, all these works analyze cases where no efficient algorithm can achieve test performance that is even *slightly* better than a random guess, although there exists a neural-network that perfectly fits the data.

While these results have great theoretical implications, we claim that they have nothing to do with understanding learnability of neural-networks on natural data. Indeed, in natural problems, even ones that are considered very challenging, achieving better-than-random performance is usually trivial. To demonstrate this, we perform the following simple experiment: we train a linear classifier on a *single* patch taken from images in some image classification task. We observe that even on a complex task such as ImageNet, a linear predictor that gets only a $3 \times 3$ patch as an input, achieves

1.5% top-5 accuracy — much better than a random guess (0.5% top-5 accuracy). Figure 1 details the results of this experiment.

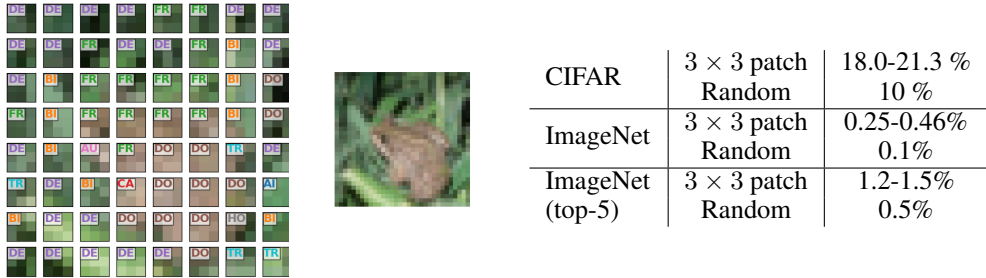

| | | |
|---|---|---|
| CIFAR | $3 \times 3$ patch | 18.0-21.3 % |
| | Random | 10 % |
| ImageNet | $3 \times 3$ patch | 0.25-0.46% |
| | Random | 0.1% |
| ImageNet (top-5) | $3 \times 3$ patch | 1.2-1.5% |
| | Random | 0.5% |

Figure 1: Performance of linear classifier on a single $3\times3$ patch from different locations in the image, trained using Adam optimizer, for 10 epochs. Left: an example image with network prediction on every patch (DE: deer, FR: frog, BI: bird, HO: horse, DO: dog). Right: mean test accuracy for patches taken from different locations in the image (worst to best) vs. a random-guess performance.

So, achieving better-than-random test performance (weak learning) is fairly easy when considering natural data, even when observing only a small fraction of the input. However, we are interested in achieving strong learning - i.e., finding a classifier with arbitrarily small error on the test data. Can we leverage our weak patch-based learners to find a strong learner? In our theoretical results (detailed in section 3.2), we show that this is indeed possible using a gradient-based layerwise optimization process, for some family of deep functions. The key property that is required for such strong learning is *local correlation*: correlation between intermediate outputs of the target function that we wish to approximate, and the target label. Here, we verify that this property holds for the ImageNet classification task. As the "target labeling function" we take a pre-trained architecture (ResNet-50), and for some layer $l$, with output of size $H \times W \times C$ (i.e., output $O \in \mathbb{R}^{H \times W \times C}$), and for some position $(i, j)$, we train a linear classifier from the $C$ channels in the $(i, j)$ position (with input $O_{i,j,:}$) to predict the label (in the ImageNet classification task). As expected, when observing higher layers, and positions closer to the center of the image, the performance of the classifier improves significantly. However, we note that even in layers very close to the input and far from the center, the accuracy of the predictor is much better than a random guess. The results of this experiments are shown in figure 2.

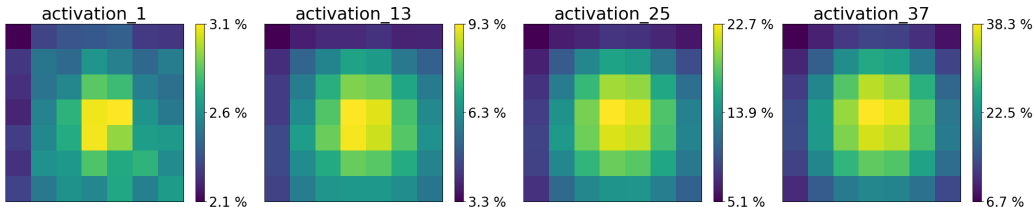

Figure 2: Top-5 accuracy of linear readout from feature-vectors in different layers of a pre-trained ResNet-50 network. The classifiers are trained with Adam optimizer, for 5 epochs, to predict the ImageNet classes. Each pixel in the image shows the top-5 accuracy of the $(i, j)$ feature-vector in the given layer, sampled in a $7 \times 7$ grid. Note that random-guess performance is $0.5\%$.

## 2    Local correlations in $k$-parities

In the previous section, we showed that in natural distributions, small parts of the input already contain some information about the target label. We start by formalizing the notion of local correlations in a well-known learning problem: the $k$-parity problem. Let $\mathcal{X} = \{\pm 1\}^n$ be the instance space and $\mathcal{Y} = \{\pm 1\}$ be the label set. In the $k$-parity problem, there is a subset $I \subseteq [n]$ of $k$ relevant bits (which are unknown to the learner), and the label is given by $f_I(\boldsymbol{x}) = \prod_{i \in I} x_i$. Namely, the

label is 1 if the number of $-1$'s among the relevant bits is even and $-1$ otherwise. The problem of learning parities was studied extensively in the literature of learning theory [17, 18, 8, 33, 10]. It is well known (e.g. [33]) that the parity problem can be expressed by a fully connected two layer network or by a depth $\log(n)$ locally connected [1] network.

It is known that when the distribution over the inputs is uniform, weakly learning $k$-parities using any noisy[2] gradient-based algorithm needs $n^{\Omega(k)}$ updates (see [33]). So, unless $k$ is constant, gradient-based algorithms cannot efficiently find a better-than-random predictor for the $k$-parity problem, under the uniform distribution. However, we observe that when the distribution is biased, finding a better-than-random predictor is trivial.

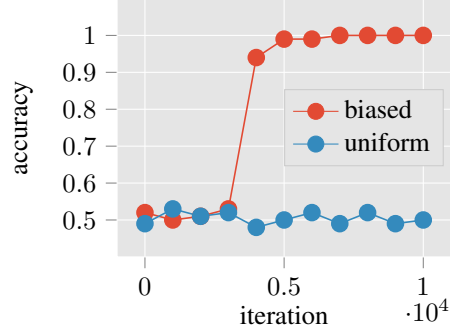

Figure 3: Training depth-two ReLU networks of size 128 with Adam, on both instances of the $k$-parity problem ($k = 5, n = 128$). The figure shows the accuracy on a test set.

To see this formally, fix some distribution $\mathcal{D}$ over $\mathcal{X}$, some subset $I \subseteq [n]$ of size $|I| = k$, and some coordinate $i \in [n]$. Denote the accuracy of $x_i$ with respect to $f_I$ by $\mathcal{C}_{\mathcal{D}}(x_i) = \mathbb{P}_{\mathcal{D}}[x_i = f_I(\boldsymbol{x})]$. Denote the bias of the target $f_I$ by $b := \mathbb{P}_{\mathcal{D}}[f_I(\boldsymbol{x}) = 1]$, and without loss of generality we assume that $b \geq 1/2$. Observe that the bias is the accuracy of the trivial predictor that always predict the label 1, and this trivial predictor is better than the "random guess" predictor (whose accuracy is always $1/2$). It follows that $\mathcal{C}_{\mathcal{D}}(x_i) > b$ if and only if the function $x \mapsto x_i$ is a non-trivial predictor of the label $f_I(\boldsymbol{x})$. Next, using

$$\mathbb{E}_{\mathcal{D}}[x_i f_I(\boldsymbol{x})] = \mathbb{P}_{\mathcal{D}}[x_i = f_I(\boldsymbol{x})] - \mathbb{P}_{\mathcal{D}}[x_i \neq f_I(\boldsymbol{x})] = \mathcal{C}_{\mathcal{D}}(x_i) - (1 - \mathcal{C}_{\mathcal{D}}(x_i)) = 2\mathcal{C}_{\mathcal{D}}(x_i) - 1,$$

using the definition $f_I(\boldsymbol{x}) = \prod_{j \in I} x_j$, and assuming that $x_j, x_{j'}$ are independent for every $j, j'$, we get that

$$\mathcal{C}_{\mathcal{D}}(x_i) = \frac{1}{2} + \frac{1}{2}\mathbb{E}_{\mathcal{D}}[x_i f_I(\boldsymbol{x})] = \frac{1}{2} + \frac{1}{2} \prod_{j \in I \setminus \{i\}} \mathbb{E}_{\mathcal{D}}[x_j]$$

So, when $\mathcal{D}$ is the uniform distribution, $\mathbb{E}_{\mathcal{D}}[x_j] = 0$ and therefore for every $i \in [n]$ we have that $\mathcal{C}_{\mathcal{D}}(x_i) = 1/2 = b$, while if $\mathcal{D}$ is a biased distribution, where every bit is 1 w.p. $\frac{1}{2} + \epsilon$, we have that $\mathbb{E}_{\mathcal{D}}[x_j] = 2\epsilon$, which yields, for every $i \in I$:

$$\mathcal{C}_{\mathcal{D}}(x_i) = \frac{1}{2} + \frac{1}{2}(2\epsilon)^{k-1} > \frac{1}{2} + \frac{1}{2}(2\epsilon)^k = b.$$

Therefore, if $k = O(\log n)$, we get that in the biased distribution, the bits of the parity weakly approximate the target label. In this case, to solve the $k$-parity problem it is enough to iterate over all the input bits, and find all the bits that predict the input with better-than-chance performance. Since all the bits outside the set $I$ are independent of the label, this gives a simple algorithm for finding the set $I$ *exactly*. That is, in this simple example, we can use the fact that weak learning is easy to find a strong learner.

The above discussion shows two distributions, the uniform one and the biased one, where the former has no local correlations between input bits and the target label and the latter has local correlations between the relevant bits and the target label. In addition, the former is not learnable by gradient-based algorithms while the latter is learnable by a simple algorithm. In the next section we will formalize this distinction by generalizing the problem, but before that, to complete the picture for parities, we observe the empirical behavior of a one hidden-layer neural network trained on the $k$-parity problem, for the uniform and biased distributions (in the biased case, the probability of every bit to be 1 is 0.6). As can be seen in figure 3, adding a slight bias to the probability of each bit dramatically affects the behavior of the network: while on the uniform distribution the training process completely fails, in the biased case it converges to a perfect solution.

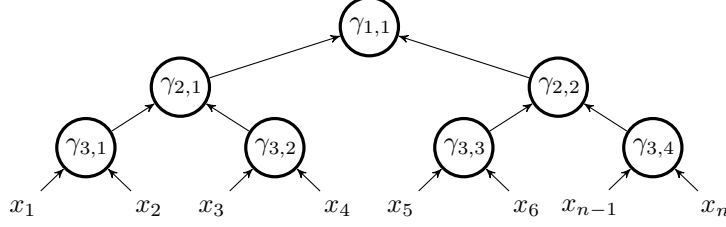

Figure 4: Tree-structured Boolean Circuit.

## 3 Learning Tree-structured Boolean Circuits

In this section we give a formal definition of the *local correlation assumption* (LCA), that is aligned with the properties of natural data we observe in section 1. We focus on the class of tree-structured Boolean circuits, which generalize the $k$-parity problem. For this class of functions, we construct a gradient-based algorithm for learning deep architectures and prove that it is a strong learner on distributions that satisfy LCA.

### 3.1 Problem Setting

We define a circuit $C$ to be a directed graph with $n$ input nodes and a single output node, where each inner node has exactly two incoming edges, and is labeled by some arbitrary Boolean function $f : \{\pm 1\}^2 \to \{\pm 1\}$, which we call a gate. For each node $v$ in $C$ we denote by $\gamma(v) \in \{f : \{\pm 1\}^2 \to \{\pm 1\}\}$ its gate. We recursively define $h_{v,C} : \{\pm 1\}^n \to \{\pm 1\}$ to be:

$$h_{v,C}(\boldsymbol{x}) = \gamma(v)\left(h_{u_1,C}(\boldsymbol{x}), h_{u_2,C}(\boldsymbol{x})\right)$$

where $u_1, u_2$ are the nodes with outcoming edges to $v$. Define $h_C = h_{o,C}$, for the output node $o$.

We study the problem of learning the target function $h_C$, when $C$ is a full binary tree, and $n = 2^d$, where $d$ is the depth of the tree. The leaves of the tree are the input bits, ordered by $x_1, \dots x_n$. We denote by $\mathcal{H}$ the family of functions that can be implemented by such tree-structured Boolean circuit. Admittedly, $\mathcal{H}$ is much smaller than the family of all circuits of depth $d$, but still gives a rather rich family of functions. For example, $\mathcal{H}$ contains all the parity functions on any $k$ bits of the input. We note that the the size of $\mathcal{H}$ grows like $6^n$, as shown in [16].

We introduce a few notations that are used in the sequel. Fix some tree-structured Boolean circuit $C$. This circuit has $d$ levels, and we denote $v_{i,j}$ the $j$-th node in the $i$-th level of the tree, and denote $\gamma_{i,j} = \gamma(v_{i,j})$. Fix some $i \in [d]$, let $n_i := 2^i$, and denote by $\Gamma_i : \{\pm 1\}^{n_i} \to \{\pm 1\}^{n_i/2}$ the function calculated by the $i$-th level of the circuit: $\Gamma_i(\boldsymbol{x}) = \left(\gamma_{i-1,1}(x_1, x_2), \dots, \gamma_{i-1,n_i/2}(x_{n_i-1}, x_{n_i})\right)$. For $i < i'$, we denote: $\Gamma_{i\dots i'} := \Gamma_i \circ \dots \circ \Gamma_{i'}$. So, the full circuit is given by $h_C(\boldsymbol{x}) = \Gamma_{1\dots d}(\boldsymbol{x})$.

As noted, our goal is to learn Boolean circuits with neural-networks. To do so, we use a network architecture that aims to imitate the Boolean circuits described above. We replace each Boolean gate with a *neural-gate*: a one hidden-layer ReLU network, with a hard-tanh[3] activation on its output. Formally, let $\sigma$ be the ReLU activation, and let $\phi$ be the hard-tanh activation, so:

$$\sigma(x) = \max(x, 0), \ \phi(x) = \max(\min(x, 1), -1)$$

Define a *neural-gate* to be a neural-network with one hidden-layer of size $k$, input dimension 2, with ReLU activation for the hidden-layer and hard-tanh for the output node. So, denote $g_{\boldsymbol{w},\boldsymbol{v}} : \mathbb{R}^2 \to \mathbb{R}$ s.t.:

$$g_{\boldsymbol{w},\boldsymbol{v}}(\boldsymbol{x}) = \phi(\sum_{l=1}^{k} v_i \sigma(\langle \boldsymbol{w}_l, \boldsymbol{x} \rangle))$$

where $\boldsymbol{w}, \boldsymbol{v}$ are the weights of the first and second layer respectively. Notice that a *neural-gate* $g_{\boldsymbol{w},\boldsymbol{v}}$ of width 4 or more can implement any Boolean gate. That is, we can replace any Boolean gate with a

neural-gate, and maintain the same expressive power. To implement the full Boolean circuit defined above, we construct a deep network of depth $d$ (the depth of the Boolean circuit), with the same structure as the Boolean circuit. We define $d$ blocks, each block has *neural-gates* with the same structure and connectivity as the Boolean circuit. A block $B_{\boldsymbol{W}^{(i)}, \boldsymbol{V}^{(i)}} : \mathbb{R}^{2^i} \to \mathbb{R}^{2^{i-1}}$, is defined by:

$$B_{\boldsymbol{W}^{(i)}, \boldsymbol{V}^{(i)}}(\boldsymbol{x}) = [g_{\boldsymbol{w}^{(i,1)}, \boldsymbol{v}^{(i,1)}}(x_1, x_2), g_{\boldsymbol{w}^{(i,2)}, \boldsymbol{v}^{(i,2)}}(x_3, x_4), \dots, g_{\boldsymbol{w}^{(i,2^{i-1})}, \boldsymbol{v}^{(i,2^{i-1})}}(x_{2^i-1}, x_{2^i})]$$

We consider the process of training neural-networks of the form $\mathcal{N}_{\mathsf{W},\mathsf{V}} = B_{\boldsymbol{W}^{(1)}, \boldsymbol{V}^{(1)}} \circ \dots \circ B_{\boldsymbol{W}^{(d)}, \boldsymbol{V}^{(d)}}$. Notice that indeed, a network $\mathcal{N}_{\mathsf{W},\mathsf{V}}$ can implement any tree-structured Boolean circuit of depth $d$. We analyze a layerwise optimization algorithm, that performs gradient updates layer-by-layer. Such approach has been recently shown to achieve performance that is comparable to end-to-end training, scaling up to the ImageNet dataset [6].

Denote by $P$ the average-pooling operator, defined by $P(x_1, \dots, x_n) = \frac{1}{n} \sum_{i=1}^{n} x_i$. To define the objective, we use the hinge-loss defined by $\ell(\hat{y}, y) = \max(1 - y\hat{y}, 0)$, and add a regularization term $R_\lambda(\hat{y}) = \lambda |1 + \hat{y}|$ to break symmetry in the optimization. We denote the overall loss on the distribution $\mathcal{D}$ and on a sample $S \subseteq \mathcal{X} \times \mathcal{Y}$ by:

$$L_{\mathcal{D}}(f) = \mathbb{E}_{(\boldsymbol{x},y) \sim \mathcal{D}} \left[ \ell(f(\boldsymbol{x}), y) + R_\lambda(f(\boldsymbol{x})) \right], \; L_S(f) = \frac{1}{|S|} \sum_{(\boldsymbol{x},y) \in S} \ell(f(\boldsymbol{x}), y) + R_\lambda(f(\boldsymbol{x}))$$

The layerwise gradient-descent algorithm for learning deep networks is described in algorithm 1.

---

**Algorithm 1** Layerwise Gradient-Descent

---

   **input**:
      Sample $S \subseteq \mathcal{X} \times \mathcal{Y}$, number of iterations $T \in \mathbb{N}$, learning rate $\eta \in \mathbb{R}$.
   Let $\mathcal{N}_d \leftarrow id$
   **for** $i = d \dots 1$ **do**
      Initialize $\boldsymbol{W}_0^{(i)}, \boldsymbol{V}_0^{(i)}$.
      **for** $t = 1 \dots T$ **do**
         Update $\boldsymbol{W}_t^{(i)} \leftarrow \boldsymbol{W}_{t-1}^{(i)} - \eta \frac{\partial}{\partial \boldsymbol{W}_{t-1}^{(i)}} L_S(P(B_{\boldsymbol{W}_{t-1}^{(i)}, \boldsymbol{V}_0^{(i)}} \circ \mathcal{N}_i))$
      **end for**
      Update $\mathcal{N}_{i-1} \leftarrow B_{\boldsymbol{W}_T^{(i)}, \boldsymbol{V}_0^{(i)}} \circ \mathcal{N}_i$
   **end for**
   Return $\mathcal{N}_0$

---

For simplicity, we assume that the second layer of every *neural-gate* is randomly initialized and fixed, such that $v \in \{\pm 1\}$. Notice that this does not limit the expressive power of the network. Algorithm 1 iteratively optimizes the output of the network's layers, starting from the bottom-most layer. For each layer, the average-pooling operator is applied to reduce the output of the layer to a single bit, and this output is optimized with respect to the target label. Note that in fact, we can equivalently optimize each *neural-gate* separately and achieve the same algorithm. However, we present a layerwise training process to conform with algorithms used in practice.

## 3.2 Main Results

Our main result shows that algorithm 1 can learn a function implemented by the circuit $C$, when running on "nice" distributions, with the local correlation property. We start by describing the distributional assumptions needed for our main results. Let $\mathcal{D}$ be some distribution over $\mathcal{X} \times \mathcal{Y}$. For some function $f : \mathcal{X} \to \mathcal{Z}$, we denote by $f(\mathcal{D})$ the distribution of $(f(\boldsymbol{x}), y)$ where $(\boldsymbol{x}, y) \sim \mathcal{D}$. Let $\mathcal{D}^{(i)}$ be the distribution $\Gamma_{(i+1)\dots d}(\mathcal{D})$. Denote by $c_{i,j}$ the correlation between the output of the $j$-th gate in the $i$-th layer and the label, so: $c_{i,j} := \mathbb{E}_{\mathcal{D}^{(i)}}[x_j y]$. Denote the influence of the $(i,j)$ gate with respect to the uniform distribution ($U$) by:

$$\mathcal{I}_{i,j} := \mathbb{P}_{\boldsymbol{x} \sim U} \left[ \Gamma_{i-1}(\boldsymbol{x}) \neq \Gamma_{i-1}(\boldsymbol{x} \oplus e_j) \right] := \mathbb{P}_{\boldsymbol{x} \sim U} \left[ \Gamma_{i-1}(\boldsymbol{x}) \neq \Gamma_{i-1}(x_1, \dots, -x_j, \dots, x_n) \right]$$

We assume w.l.o.g. that for every $i, j$ such that $\mathcal{I}_{i,j} = 0$, the $(i,j)$ gate is constant $\gamma_{i,j} \equiv 1$. Since the output of this gate has no influence on the output of the circuit, we can choose it freely without changing the target function. Again w.l.o.g., we assume $\mathbb{E}_{\mathcal{D}}[y] \geq 0$. Indeed, if this does not hold, we can observe the same distribution with swapped labels. Now, our main assumption is the following:

**Assumption 1.** *(LCA) There exists some $\Delta \in (0,1)$ such that for every layer $i \in [d]$ and for every gate $j \in [2^i]$ with $\mathcal{I}_{i,j} \neq 0$, the value of $c_{i,j}$ satisfies $|c_{i,j}| > \mathbb{E}_{\mathcal{D}}[y] + \Delta$.*

Essentially, this assumption requires that the output of every gate in the circuit will "explain" the label slightly better than simply observing the bias between positive and negative examples. Clearly, gates that have no influence on the target function never satisfy this property, so we require it only for influencing gates. Recall that in section 1 we showed that a similar property indeed holds for natural data. In section 3.3 we discuss examples of distributions where this assumption typically holds.

Now, we start by considering the case of non-degenerate product distributions:

**Definition 1.** *A distribution $\mathcal{D}$ is a $\Delta$-non-degenerate product distribution, if the following holds:*

- *For every $j \neq j'$, the variables $x_j$ and $x_{j'}$ are independent, for $(\boldsymbol{x}, y) \sim \mathcal{D}$.*

- *For every $j$, we have $\mathbb{P}_{(\boldsymbol{x},y)\sim\mathcal{D}}[x_j = 1] \in (\Delta, 1 - \Delta)$.*

Our first result shows that for non-degenerate product distributions satisfying LCA, our algorithm returns a network with zero loss w.h.p., with polynomial sample complexity and run-time:

**Theorem 1.** *Fix $\delta, \Delta \in (0, \frac{1}{2})$ and integer $n = 2^d$. Let $k \geq \log^{-1}(\frac{4}{3})\log(\frac{2nd}{\delta})$, $\eta \leq \frac{1}{32k}$ and $\lambda = \mathbb{E}[y] + \frac{\Delta}{4}$. Fix some $h \in \mathcal{H}$, and let $\mathcal{D}$ be a $\Delta$-non-degnerate product distribution labeled by $h$ s.t. $\mathcal{D}$ satisfies assumption 1 (LCA) with parameter $\Delta$. Assume we sample $S \sim \mathcal{D}$, with $|S| > \frac{2^{15}}{\Delta^6}\log(\frac{8nd}{\delta})$. Then, w.p. $\geq 1 - \delta$, when running algorithm 1 with initialization of $\boldsymbol{W}$ s.t. $\left\|\boldsymbol{W}_0^{(i)}\right\|_{\max} \leq \frac{1}{4\sqrt{2k}}$ on the sample $S$, the algorithm returns a function $\mathcal{N}_0$ s.t. $\mathbb{P}_{(\boldsymbol{x},y)\sim\mathcal{D}}[\mathcal{N}_0(\boldsymbol{x}) \neq y] = 0$, when running $T > \frac{24n}{\sqrt{2}\eta\Delta^3}$ steps per layer.*

In fact, we can show this result for a larger family of distributions, going beyond product distributions. First, we show that a non-degenerate product distribution $\mathcal{D}$ that satisfies LCA, satisfies the following properties:

**Property 1.** *There exists some fixed $\Delta \in (0,1)$ such that for every layer $i \in [d]$ and for every gate $j \in [2^i]$, the output of the $j$-th gate in the $i$-th layer satisfies one of the following:*

- *The value of the gate $j$ is independent of the label $y$, and its influence is zero: $\mathcal{I}_{i,j} = 0$.*

- *The value of $c_{i,j}$ satisfies $|c_{i,j}| > \mathbb{E}_{\mathcal{D}}[y] + \Delta$.*

**Property 2.** *For every layer $i \in [d]$, and every gate $j \in [2^{i-1}]$, the value of $(x_{2j-1}, x_{2j})$ (i.e, the input to the $j$-th gate of layer $i - 1$) is independent of the label $y$ given the output of the $j$-th gate:*

$$\mathbb{P}_{(x,y)\sim\mathcal{D}^{(i)}}[(x_{2j-1}, x_{2j}) = \boldsymbol{p}, y = y'|\gamma_{i-1,j}(x_{2j-1}, x_{2j})]$$
$$= \mathbb{P}_{(\boldsymbol{x},y)\sim\mathcal{D}^{(i)}}[(x_{2j-1}, x_{2j}) = \boldsymbol{p}|\gamma_{i-1,j}(x_{2j-1}, x_{2j})] \cdot \mathbb{P}_{(\boldsymbol{x},y)\sim\mathcal{D}^{(i)}}[y = y'|\gamma_{i-1,j}(x_{2j-1}, x_{2j})]$$

**Property 3.** *There exists some fixed $\epsilon \in (0,1)$ such that for every layer $i \in [d]$, for every gate $j \in [2^{i-1}]$ and for every $\boldsymbol{p} \in \{\pm 1\}^2$ such that $\mathbb{P}_{(\boldsymbol{x},y)\sim\mathcal{D}^{(i)}}[(x_{2j-1}, x_{2j}) = \boldsymbol{p}] > 0$, it holds that: $\mathbb{P}_{(\boldsymbol{x},y)\sim\mathcal{D}^{(i)}}[(x_{2j-1}, x_{2j}) = \boldsymbol{p}] \geq \epsilon$.*

From the following lemma, these properties generalize the case of product distributions with LCA:

**Lemma 1.** *Any $\Delta$-non-degenerate product distribution $\mathcal{D}$ satisfying assumption 1 (LCA), satisfies properties 1-3, with $\epsilon = \frac{\Delta^2}{4}$.*

Notice that properties 1, 2 and 3 may hold for distributions that are not product distributions (as we show in the next section). Specifically, property 2 is a very common assumption in the field of Graphical Models (see [23]). As in Theorem 1, given a distribution satisfying properties 1-3, we show that with high probability, algorithm 1 returns a function with zero loss, with sample complexity and run-time polynomial in the dimension $n$:

**Theorem 2.** *Fix $\delta, \Delta, \epsilon \in (0, \frac{1}{2})$ and integer $n = 2^d$. Let $k \geq \log^{-1}(\frac{4}{3})\log(\frac{2nd}{\delta})$, $\eta \leq \frac{1}{32k}$ and $\lambda = \mathbb{E}[y] + \frac{\Delta}{4}$. Fix some $h \in \mathcal{H}$, and let $\mathcal{D}$ be a distribution labeled by $h$ which satisfies properties 1-3 with $\Delta, \epsilon$. Assume we sample $S \sim \mathcal{D}$, with $|S| > \frac{2^{11}}{\epsilon^2\Delta^2}\log(\frac{8nd}{\delta})$. Then, w.p. $\geq 1 - \delta$,*

*when running algorithm 1 with initialization of $\boldsymbol{W}$ s.t. $\left\| \boldsymbol{W}_0^{(i)} \right\|_{\max} \leq \frac{1}{4\sqrt{2}k}$ on the sample $S$, the algorithm returns a function s.t. $\mathbb{P}_{(\boldsymbol{x},y)\sim\mathcal{D}}\left[\mathcal{N}_0(\boldsymbol{x}) \neq y\right] = 0$, running $T > \frac{6n}{\sqrt{2}\eta\epsilon\Delta}$ steps per layer.*

We give the full proof of the theorems in the appendix, and give a sketch of the argument here. First, note that Theorem 1 follows from Theorem 2 and Lemma 1. To prove Theorem 2, observe that the input to the $(i,j)$-th *neural-gate* is a pattern of two bits. The target gate (the $(i,j)$-th gate in the circuit $C$) identifies each of the four possible patterns with a single output bit. For example, if the gate is OR, then the patterns $\{(1,1),(-1,1),(1,-1)\}$ get the value 1, and the pattern $(-1,-1)$ gets the value $-1$. Fix some pattern $\boldsymbol{p} \in \{\pm1\}^2$, and assume that the output of the $(i,j)$-th gate on the pattern $\boldsymbol{p}$ is 1. Since we assume the output of the gate is correlated with the label, the loss function draws the output of the *neural-gate* on the pattern $\boldsymbol{p}$ toward the *correlation* of the gate. In the case where the output of the gate on $\boldsymbol{p}$ is $-1$, the output of the *neural-gate* is drawn to the opposite sign of the *correlation*. All in all, the optimization separates the patterns that evaluate to 1 from the patterns that evaluate to $-1$. In other words, the *neural-gate* learns to implement the target gate. This way, we can show that the network recovers all the influencing gates, so at the end of the optimization process the network implements the circuit.

Observe that when there is no correlation, the above argument fails immediately. Without correlation, the output of the *neural-gate* is drawn towards a constant value for all the input patterns, regardless of the value of the gate. If the gate is not influencing the target function (i.e. $\mathcal{I}_{i,j} = 0$), then this clearly doesn't affect the overall behavior. However, if there exists some influencing gate with no correlation to the label, then the output of the *neural-gate* will be constant on all its input patterns. Hence, the algorithm will fail to recover the target function. This shows that LCA is in fact critical for the success of the algorithm.

### 3.3 Distributions that Satisfy LCA

We showed that algorithm 1 can learn tree-structured Boolean circuits in polynomial run-time and sample complexity. These results require some non-trivial distributional assumptions. We now analyze specific families of distributions, and show that they satisfy the above assumptions. First, we study the problem of learning a parity function on $\log n$ bits of the input, when the underlying distribution is a product distribution (as discussed in section 2). In the $(\log n)$-parity problem, we show that in fact any product distributions far enough from the uniform distribution satisfies assumption 1, hence algorithm 1 learns such distributions:

**Theorem 3.** *Fix some constant $\xi \in (0, 1/4)$. Let $\mathcal{D}$ be a product distribution with $\mathbb{P}_{\mathcal{D}}[x_j = 1] \in (\xi, \frac{1}{2}-\xi)\cup(\frac{1}{2}+\xi, 1-\xi)$ for every $j$, with the target function being a $(\log n)$-parity (i.e., $k = \log n$). Then, when running algorithm 1 as described in Theorem 2, with probability at least $1 - \delta$ the algorithm returns a circuit that implements the true target function $f_I$, with run-time and sample complexity polynomial in $n$.*

Next, we study distributions given by a generative model. We show that for every circuit with gates AND/OR/NOT, there exists a distribution that satisfies properties 1-3, so algorithm 1 can learn any such circuit exactly. For every AND/OR circuit, we define a generative distribution as follows: we start by sampling a label for the example. Then iteratively, for every gate, we sample uniformly at random a pattern from all the patterns that give the correct output. For example, if the label is 1 and the topmost gate is OR, we sample a pattern uniformly from $\{(1,1),(1,-1),(-1,1)\}$. The sampled pattern determines what should be the output of the second topmost layer. For every gate in this layer, we sample again a pattern that will result in the correct output. We continue in this fashion until reaching the bottom-most layer, which defines the observed example. We show that such a distribution satisfies properties 1-3, and so algorithm 1 exactly recovers the circuit:

**Theorem 4.** *With the assumptions and notations of Theorem 2, for every circuit $C$ with gates in $\{\wedge, \vee, \neg\wedge, \neg\vee\}$, there exists a distribution $\mathcal{D}$ such that when running algorithm 1 on a sample from $\mathcal{D}$, the algorithm returns $h_C$ with probability $1 - \delta$, in polynomial run-time and sample complexity.*

Note that the fact that for every circuit there exists a distribution that can be *approximated* by the algorithm is trivial: simply take a distribution that is concentrated on a single positive example, and approximating the target function on such distribution is achieved by a classifier that always returns a positive prediction. However, showing that there exists a distribution on which algorithm 1 *exactly* recovers the circuit, is certainly non-trivial.

# 4 Related Work

In recent years, the success of neural-networks has inspired an ongoing theoretical research, trying to explain empirical observations about their behavior. Some theoretical works show failure cases of neural-networks. Other works give different guarantees on various learning algorithms for neural-networks. In this section, we cover the main works that are relevant to our paper. **Learning neural-networks with gradient-descent**. Recently, a large number of papers have provided positive results on learning neural-networks with gradient-descent. Generally speaking, most of these works show that over-parametrized neural-networks, deep or shallow, achieve performance that is competitive with kernel methods (or kernel-SVM). Daniely [14] shows that SGD learns the conjugate kernel associated with the architecture of the network, for a wide enough neural-network. The work of Brutzkus & Globerson [13] shows that SGD learns a neural-network with good generalization, when the target function is linear. A growing number of works show that for a specific kernel induced by the network activation, called the Neural Tangent Kernel (NTK), gradient-descent learns over-parametrized networks, for target functions with small norm in the reproducing kernel Hilbert space (see the works of [19, 35, 30, 2, 3, 31, 5, 15, 27, 24]). While these results show that learning neural-networks with gradient-descent is not hopeless, they are in some sense disappointing — in practice, neural-networks achieve performance that are far better than kernel methods, a fact that is not explained by these works. A few results do discuss success cases of gradient-descent that go beyond the kernel-based analysis [11, 12, 1, 36]. However, these works still focus on very simple cases, such as learning a single neuron, or learning shallow neural-networks in restricted settings. We deal with learning deep functions, going beyond the reduction to linear models.

**Gradient-based optimization algorithms**. Deep networks are optimized using an access to the objective function (population loss) through a stochastic gradient oracle. The common algorithms that are used in practice are variants of end-to-end stochastic gradient descent, where gradients are calculated by backpropagation. However, the focus of this paper is not on a specific algorithm but on the question of what can be done with an access to a stochastic gradient oracle. In particular, for simplicity of the analysis (as was also done in [4, 28]), we analyze the behavior of layerwise optimization — optimizing one layer at a time. Recent works [6, 7] have shown that layerwise training achieves performance that is competitive with the standard end-to-end approach, scaling up to the ImageNet dataset.

**Learning Boolean Circuits**. The problem of learning Boolean circuits has been studied in the classical literature of theoretical machine learning. The work of Kearns et al. [22] gives various positive and negative results on the learnability of Boolean Formulas, including Boolean circuits. The work of Linial et al. [25] introduces an algorithm that learns a constant-depth circuit in quasi-polynomial time. Another work by Kalai [21] discusses various properties of learning Boolean formulas and Boolean circuits. The problem of learning tree-structured Boolean circuits, also known as read-once formulas, has also been studies in the PAC learning literature [32]. Our work differs from the above in various aspects. Our main focus is learning deep neural-networks with gradient descent, where the target function is implemented by a Boolean circuit, and we do not aim to study the learnability of Boolean circuits in general. Furthermore, we consider Boolean circuits where a gate can take any Boolean function, and not only AND/OR/NOT, as is often considered in the literature of Boolean circuits. On the other hand, we restrict ourselves to the problem of learning circuits with a fixed structure of full binary trees.

**Learning and Smoothed Analysis** We note that another popular approach for overcoming worst-case hardness results is through the technique of smoothed analysis [34]. This setting was used to show learnability of DNFs and decision trees, in a notable work by Kalai et al. [20]. While their results are similar in spirit to ours, there are some crucial differences. First, their results apply only to product distributions, while our results apply to a larger family of distributions. We are unaware of any immediate extension of the smoothed analysis setting beyond the case of product distributions. Second, the results of [20] apply to DNFs and decision trees, while we focus on learning tree-structured Boolean circuits.

## Broader Impact

The revolution of Deep Learning has allowed unprecedented progress in almost any field of Artificial Intelligence. However, this progress is mainly guided by empirical experiments, where improvements are typically achieved by trial-and-error. This is due to the fact that current learning theory is not applicable to practical settings: negative theoretical results take a worst-case analysis, which is irrelevant to practical problems, while positive results analyze very simplistic cases.

The goal of this paper is to identify what are the correct distributional assumptions that are required for constructing theoretical results that are relevant to natural problems. We suggested the *local correlation assumption* (LCA), an assumption that is likely to hold on natural data, and allows us to prove some non-trivial theoretical results. We showed that on the task of learning tree-structured Boolean circuits, the existence of *local correlations* between the gates and the target label allows layerwise gradient-descent to learn the target circuit. We believe this work can help the research community to come up with a theory of deep networks that is applicable in practice, and encourage theory-guided progress in solving real-world problems.

## Acknowledgments and Disclosure of Funding

This research is supported by the European Research Council (TheoryDL project).

## Footnotes

[1]i.e. every two adjacent neurons are only connected to one neuron in the upper layer.

[2]Even an extremely small amount of noise, below the machine precision.

[3]We chose to use the hard-tanh activation over the more popular tanh activation since it simplifies our theoretical analysis. However, we believe the same results can be given for the tanh activation.

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
