[Supplementary Material]

# A Proof of Theorem 2

To prove Theorem 2, we observe the behavior of the algorithm on the $i$-th layer. Let $\psi : \{\pm 1\}^{n_i/2} \to \{\pm 1\}^{n_i/2}$ be some mapping such that $\psi(\boldsymbol{x}) = (\xi_1 \cdot x_1, \ldots, \xi_{n_i/2} \cdot x_{n_i/2})$ for $\xi_1, \ldots, \xi_{n_i/2} \in \{\pm 1\}$. We also define $\varphi_i : \{\pm 1\}^{n_i/2} \to \{\pm 1\}^{n_i/2}$ such that:

$$\varphi_i(\boldsymbol{z}) = (\nu_1 z_1, \ldots, \nu_{n_i/2} z_{n_i/2})$$

where $\nu_j := \begin{cases} \mathrm{sign}(c_{i-1,j}) & c_{i-1,j} \neq 0 \\ -1 & \mathcal{I}_{i-1,j} = 0 \end{cases}$

We can ignore examples that appear with probability zero. For this, we define the support of $\mathcal{D}$ by $\mathcal{X}' = \{\boldsymbol{x}' \in \mathcal{X} : \mathbb{P}_{(\boldsymbol{x},y)\sim\mathcal{D}}[\boldsymbol{x} = \boldsymbol{x}'] > 0\}$.

We have the following important result, which we prove in the sequel:

**Lemma 2.** *Assume we initialize $\boldsymbol{w}_l^{(0)}$ such that $\left\|\boldsymbol{w}_l^{(0)}\right\| \leq \frac{1}{4k}$. Fix $\delta > 0$. Assume we sample $S \sim \mathcal{D}$, with $|S| > \frac{2^{11}}{\epsilon^2 \Delta^2} \log(\frac{8n_i}{\delta})$. Assume that $k \geq \log^{-1}(\frac{4}{3}) \log(\frac{8n_i}{\delta})$, and that $\eta \leq \frac{n_i}{32k}$. Let $\Psi : \mathcal{X} \to [-1,1]^{n_i/2}$ such that for every $\boldsymbol{x} \in \mathcal{X}'$ we have $\Psi(\boldsymbol{x}) = \psi \circ \Gamma_{(i+1)\ldots d}(\boldsymbol{x})$ for some $\psi$ as defined above. Assume we perform the following updates:*

$$\boldsymbol{W}_t^{(i)} \leftarrow \boldsymbol{W}_{t-1}^{(i)} - \eta \frac{\partial}{\partial \boldsymbol{W}_{t-1}^{(i)}} L_{\Psi(S)}(P(B_{\boldsymbol{W}_{t-1}^{(i)}, \boldsymbol{V}_0^{(i)}}))$$

*Then with probability at least $1 - \delta$, for $t > \frac{6n_i}{\sqrt{2}\eta\epsilon\Delta}$ we have: $B_{\boldsymbol{W}_t^{(i)}, \boldsymbol{V}_0^{(i)}}(\boldsymbol{x}) = \varphi_i \circ \Gamma_i \circ \psi(\boldsymbol{x})$ for every $\boldsymbol{x} \in \Psi(\mathcal{X}')$.*

Given this result, we can prove the main theorem:

*Proof.* of Theorem 2. Fix $\delta' = \frac{\delta}{d}$. We show that for every $i \in [d]$, w.p at least $1 - (d-i+1)\delta'$, after the $i$-th step of the algorithm we have $\mathcal{N}_{i-1}(\boldsymbol{x}) = \varphi_i \circ \Gamma_{i\ldots d}(\boldsymbol{x})$ for every $\boldsymbol{x} \in \mathcal{X}'$. By induction on $i$:

- For $i = d$, we get the required using Lemma 2 with $\psi, \Psi = id$.

- Assume the above holds for $i$, and we show it for $i - 1$. By the assumption, w.p at least $1 - (d-i+1)\delta'$ we have $\mathcal{N}_{i-1}(\boldsymbol{x}) = \varphi_i \circ \Gamma_{i\ldots d}(\boldsymbol{x})$ for every $\boldsymbol{x} \in \mathcal{X}'$. Observe that:

$$\frac{\partial L_S}{\partial \boldsymbol{W}_t^{(i-1)}}(P(B_{\boldsymbol{W}_{t-1}^{(i-1)}, \boldsymbol{V}_0^{(i-1)}} \circ \mathcal{N}_{i-1})) = \frac{\partial L_{\mathcal{N}_{i-1}(S)}}{\partial \boldsymbol{W}_t^{(i-1)}}(P(B_{\boldsymbol{W}_t^{(i-1)}, \boldsymbol{V}_0^{(i-1)}}))$$

So using Lemma 2 with $\psi = \varphi_i$, $\Psi = \mathcal{N}_{i-1}$ we get that w.p at least $1 - \delta'$ we have $B_{\boldsymbol{W}_T^{(i-1)}, \boldsymbol{V}_0^{(i-1)}}(\boldsymbol{x}) = \varphi_{i-1} \circ \Gamma_{i-1} \circ \varphi_i(\boldsymbol{x})$ for every $\boldsymbol{x} \in \mathcal{X}'$. In this case, since $\varphi_i \circ \varphi_i = id$, we get that for every $\boldsymbol{x} \in \mathcal{X}'$:

$$\begin{aligned} \mathcal{N}_{i-2}(\boldsymbol{x}) &= B_{\boldsymbol{W}_T^{(i-1)}, \boldsymbol{V}_0^{(i-1)}} \circ \mathcal{N}_{i-1}(\boldsymbol{x}) \\ &= (\varphi_{i-1} \circ \Gamma_{i-1} \circ \varphi_i) \circ (\varphi_i \circ \Gamma_{i\ldots d})(\boldsymbol{x}) = \varphi_{i-1} \circ \Gamma_{(i-1)\ldots d}(\boldsymbol{x}) \end{aligned}$$

and using the union bound gives the required.

Notice that $\varphi_1 = id$: by definition of $\mathcal{D}^{(0)} = \Gamma_{1\ldots d}(\mathcal{D})$, for $(\boldsymbol{z}, y) \sim \mathcal{D}^{(0)}$ we have $\boldsymbol{z} = \Gamma_{1\ldots d}(\boldsymbol{x})$ and also $y = \Gamma_{1\ldots d}(\boldsymbol{x})$ for $(\boldsymbol{x}, y) \sim \mathcal{D}$. Therefore, we have $c_{0,1} = \mathbb{E}_{(x,y)\sim\mathcal{D}^{(0)}}[xy] = 1$, and therefore $\varphi_i(z) = \mathrm{sign}(c_{0,1})z = z$. Now, choosing $i = 1$, the above result shows that with probability at least $1 - \delta$, the algorithm returns $\mathcal{N}_0$ such that $\mathcal{N}_0(\boldsymbol{x}) = \varphi_1 \circ \Gamma_1 \circ \cdots \circ \Gamma_d(\boldsymbol{x}) = h_C(\boldsymbol{x})$ for every $\boldsymbol{x} \in \mathcal{X}'$. $\qquad\square$

In the rest of this section we prove Lemma 2. Fix some $i \in [d]$ and let $j \in [n_i/2]$. With slight abuse of notation, we denote by $\boldsymbol{w}^{(t)}$ the value of the weight $\boldsymbol{w}^{(i,j)}$ at iteration $t$, and denote $\boldsymbol{v} := \boldsymbol{v}^{(i,j)}$ and $g_t := g_{\boldsymbol{w}^{(t)}, \boldsymbol{v}}$. Recall that we defined $\psi(\boldsymbol{x}) = (\xi_1 \cdot x_1, \ldots, \xi_{n_i} \cdot x_{n_i})$ for $\xi_1 \ldots \xi_{n_i} \in \{\pm 1\}$. Let

$\gamma := \gamma_{i-1,j}$, and let $\widetilde{\gamma}$ such that $\widetilde{\gamma}(x_1, x_2) = \gamma(\xi_{2j-1} \cdot x_1, \xi_{2j} \cdot x_2)$. For every $\boldsymbol{p} \in \{\pm 1\}^2$, denote $\widetilde{\boldsymbol{p}} := (\xi_{2j-1}p_1, \xi_{2j}p_2)$, so we have $\gamma(\widetilde{\boldsymbol{p}}) = \widetilde{\gamma}(\boldsymbol{p})$. Now, we care only about patterns $\boldsymbol{p}$ that have positive probability to appear as input to the gate $(i - 1, j)$. So, we define our pattern support by:

$$\mathcal{P} = \{\boldsymbol{p} \in \{\pm 1\}^2 \ : \ \mathbb{P}_{(\boldsymbol{x},y) \sim \Psi(\mathcal{D})}\left[(x_{2j-1}, x_{2j}) = \boldsymbol{p}\right] > 0\}$$

Finally, if the gate $\gamma_{i-1,j}$ has no influence on the target function (i.e., if $\mathcal{I}_{i-1,j} = 0$), we can choose it arbitrarily without affecting the output of the circuit. So, w.l.o.g. we assume in this case that $\widetilde{\gamma} \equiv 1$. We start by observing the behavior of the gradient with respect to some pattern $\boldsymbol{p} \in \mathcal{P}$:

**Lemma 3.** *Fix some $\boldsymbol{p} \in \mathcal{P}$. For every $l \in [k]$ such that $\langle \boldsymbol{w}_l^{(t)}, \boldsymbol{p} \rangle > 0$ and $g_t(\boldsymbol{p}) \in (-1, 1)$, the following holds:*

$$-\widetilde{\gamma}(\boldsymbol{p})v_l\nu_j\langle \frac{\partial L_{\Psi(\mathcal{D})}}{\partial \boldsymbol{w}_l^{(t)}}, \boldsymbol{p} \rangle > \frac{\epsilon}{n_i}\Delta$$

*Proof.* Observe the following:

$$\frac{\partial L_{\Psi(\mathcal{D})}}{\partial \boldsymbol{w}_l^{(t)}}(P(B_{\boldsymbol{W}^{(i)}, \boldsymbol{V}^{(i)}}))$$

$$= \mathbb{E}_{(\boldsymbol{x},y) \sim \Psi(\mathcal{D})}\left[\ell'(P(B_{\boldsymbol{W}^{(i)}, \boldsymbol{V}^{(i)}})(\boldsymbol{x})) \cdot \frac{\partial}{\partial \boldsymbol{w}_l^{(t)}} \frac{2}{n_i} \sum_{j'=1}^{n_i/2} g_{\boldsymbol{w}^{(i,j')}, \boldsymbol{v}^{(i,j')}}(x_{2j'-1}, x_{2j'})\right]$$

$$+ \mathbb{E}_{(\boldsymbol{x},y) \sim \Psi(\mathcal{D})}\left[R'_\lambda(P(B_{\boldsymbol{W}^{(i)}, \boldsymbol{V}^{(i)}})(\boldsymbol{x})) \cdot \frac{\partial}{\partial \boldsymbol{w}_l^{(t)}} \frac{2}{n_i} \sum_{j'=1}^{n_i/2} g_{\boldsymbol{w}^{(i,j')}, \boldsymbol{v}^{(i,j')}}(x_{2j'-1}, x_{2j'})\right]$$

$$= \frac{2}{n_i}\mathbb{E}_{\Psi(\mathcal{D})}\left[(\lambda - y)\frac{\partial}{\partial \boldsymbol{w}_l^{(t)}} g_t(x_{2j-1}, x_{2j})\right]$$

$$= \frac{2}{n_i}\mathbb{E}_{\Psi(\mathcal{D})}\left[(\lambda - y)v_l\mathbf{1}\{g_t(x_{2j-1}, x_{2j}) \in (-1, 1)\} \cdot \mathbf{1}\{\langle \boldsymbol{w}_l^{(t)}, (x_{2j-1}, x_{2j})\rangle > 0\} \cdot (x_{2j-1}, x_{2j})\right]$$

We use the fact that $\ell'(P(B_{\boldsymbol{W}^{(i)}, \boldsymbol{V}^{(i)}})(\boldsymbol{x})) = -y$, unless $P(B_{\boldsymbol{W}^{(i)}, \boldsymbol{V}^{(i)}})(\boldsymbol{x}) \in \{\pm 1\}$, in which case $g_t(x_{2j-1}, x_{2j}) \in \{\pm 1\}$, so $\frac{\partial}{\partial \boldsymbol{w}_l^{(t)}} g_t(x_{2j-1}, x_{2j}) = 0$. Similarly, unless $\frac{\partial}{\partial \boldsymbol{w}_l^{(t)}} g_t(x_{2j-1}, x_{2j}) = 0$, we get that $R'_\lambda(P(B_{\boldsymbol{W}^{(i)}, \boldsymbol{V}^{(i)}})(\boldsymbol{x})) = \lambda$. Fix some $\boldsymbol{p} \in \{\pm 1\}^2$ such that $\langle \boldsymbol{w}_l^{(t)}, \boldsymbol{p} \rangle > 0$. Note that for every $\boldsymbol{p} \neq \boldsymbol{p}' \in \{\pm 1\}^2$ we have either $\langle \boldsymbol{p}, \boldsymbol{p}' \rangle = 0$, or $\boldsymbol{p} = -\boldsymbol{p}'$ in which case $\langle \boldsymbol{w}_l^{(t)}, \boldsymbol{p}' \rangle < 0$. Therefore, we get the following:

$$\langle \frac{\partial L_{\Psi(\mathcal{D})}}{\partial \boldsymbol{w}_l^{(t)}}, \boldsymbol{p} \rangle$$

$$= \frac{2}{n_i}\mathbb{E}_{\Psi(\mathcal{D})}\left[(\lambda - y)v_l\mathbf{1}\{g_t(x_{2j-1}, x_{2j}) \in (-1, 1)\} \cdot \mathbf{1}\{\langle \boldsymbol{w}_l^{(t)}, (x_{2j-1}, x_{2j})\rangle \geq 0\} \cdot \langle (x_{2j-1}, x_{2j}), \boldsymbol{p} \rangle\right]$$

$$= \frac{2}{n_i}\mathbb{E}_{\Psi(\mathcal{D})}\left[(\lambda - y)v_l\mathbf{1}\{g_t(x_{2j-1}, x_{2j}) \in (-1, 1)\} \cdot \mathbf{1}\{(x_{2j-1}, x_{2j}) = \boldsymbol{p}\} \|\boldsymbol{p}\|^2\right]$$

Denote $q_{\boldsymbol{p}} := \mathbb{P}_{(\boldsymbol{x},y) \sim \mathcal{D}^{(i)}}\left[(x_{2j-1}, x_{2j}) = \boldsymbol{p}|\gamma(x_{2j-1}, x_{2j}) = \gamma(\boldsymbol{p})\right]$. Using property 2, we have:

$$\mathbb{P}_{(\boldsymbol{x},y) \sim \mathcal{D}^{(i)}}\left[(x_{2j-1}, x_{2j}) = \boldsymbol{p}, y = y'\right]$$

$$= \mathbb{P}_{(\boldsymbol{x},y) \sim \mathcal{D}^{(i)}}\left[(x_{2j-1}, x_{2j}) = \boldsymbol{p}, y = y', \gamma(x_{2j-1}, x_{2j}) = \gamma(\boldsymbol{p})\right]$$

$$= \mathbb{P}_{(\boldsymbol{x},y) \sim \mathcal{D}^{(i)}}\left[(x_{2j-1}, x_{2j}) = \boldsymbol{p}, y = y'|\gamma(x_{2j-1}, x_{2j}) = \gamma(\boldsymbol{p})\right]\mathbb{P}_{(\boldsymbol{x},y) \sim \mathcal{D}^{(i)}}\left[\gamma(x_{2j-1}, x_{2j}) = \gamma(\boldsymbol{p})\right]$$

$$= q_{\boldsymbol{p}}\mathbb{P}_{(\boldsymbol{x},y) \sim \mathcal{D}^{(i)}}\left[\gamma(x_{2j-1}, x_{2j}) = \gamma(\boldsymbol{p}), y = y'\right]$$

$$= q_{\boldsymbol{p}}\mathbb{P}_{(\boldsymbol{z},y) \sim \mathcal{D}^{(i-1)}}\left[z_j = \gamma(\boldsymbol{p}), y = y'\right]$$

And therefore:

$$\mathbb{E}_{(\boldsymbol{x},y)\sim\mathcal{D}^{(i)}}\left[y\mathbf{1}\{(x_{2j-1},x_{2j})=\boldsymbol{p}\}\right] = \sum_{y'\in\{\pm1\}} y'\mathbb{P}_{(\boldsymbol{x},y)\sim\mathcal{D}^{(i)}}\left[(x_{2j-1},x_{2j})=\boldsymbol{p},y=y'\right]$$

$$= q_{\boldsymbol{p}}\sum_{y'\in\{\pm1\}} y'\mathbb{P}_{(\boldsymbol{z},y)\sim\mathcal{D}^{(i-1)}}\left[z_j=\gamma(\boldsymbol{p}),y=y'\right]$$

$$= q_{\boldsymbol{p}}\mathbb{E}_{(\boldsymbol{z},y)\sim\mathcal{D}^{(i-1)}}\left[y\mathbf{1}\{z_j=\gamma(\boldsymbol{p})\}\right]$$

Assuming $g_t(\boldsymbol{p})\in(-1,1)$, using the above we get:

$$\langle\frac{\partial L_{\Psi(\mathcal{D})}}{\partial\boldsymbol{w}_l^{(t)}},\boldsymbol{p}\rangle = \frac{4v_l}{n_i}\mathbb{E}_{(\boldsymbol{x},y)\sim\Psi(\mathcal{D})}\left[(\lambda-y)\mathbf{1}\{(x_{2j-1},x_{2j})=\boldsymbol{p}\}\right]$$

$$= \frac{4v_l}{n_i}\mathbb{E}_{(\boldsymbol{x},y)\sim\mathcal{D}^{(i)}}\left[(\lambda-y)\mathbf{1}\{(\xi_{2j-1}x_{2j-1},\xi_{2j}x_{2j})=\boldsymbol{p}\}\right]$$

$$= \frac{4v_l}{n_i}\mathbb{E}_{(\boldsymbol{x},y)\sim\mathcal{D}^{(i)}}\left[(\lambda-y)\mathbf{1}\{(x_{2j-1},x_{2j})=\widetilde{\boldsymbol{p}}\}\right]$$

$$= \frac{4v_l q_{\widetilde{\boldsymbol{p}}}}{n_i}\mathbb{E}_{(\boldsymbol{z},y)\sim\mathcal{D}^{(i-1)}}\left[(\lambda-y)\mathbf{1}\{z_j=\widetilde{\gamma}(\boldsymbol{p})\}\right]$$

Now, we have the following cases:

- If $\mathcal{I}_{i-1,j}=0$, then by property 1 $z_j$ and $y$ are independent, so:

$$\langle\frac{\partial L_{\Psi(\mathcal{D})}}{\partial\boldsymbol{w}_l^{(t)}},\boldsymbol{p}\rangle = \frac{4v_l q_{\widetilde{\boldsymbol{p}}}}{n_i}\mathbb{E}_{(\boldsymbol{z},y)\sim\mathcal{D}^{(i-1)}}\left[(\lambda-y)\mathbf{1}\{z_j=\widetilde{\gamma}(\boldsymbol{p})\}\right]$$

$$= \frac{4v_l q_{\widetilde{\boldsymbol{p}}}}{n_i}\mathbb{E}_{(\boldsymbol{z},y)\sim\mathcal{D}^{(i-1)}}\left[(\lambda-y)\right]\mathbb{P}_{(\boldsymbol{z},y)\sim\mathcal{D}^{(i-1)}}\left[z_j=\widetilde{\gamma}(\boldsymbol{p})\right]$$

$$= \frac{4v_l}{n_i}(\lambda-\mathbb{E}_{(\boldsymbol{z},y)\sim\mathcal{D}^{(i-1)}}\left[y\right])\mathbb{P}_{(\boldsymbol{x},y)\sim\mathcal{D}^{(i)}}\left[(x_{2j-1},x_{2j})=\widetilde{\boldsymbol{p}}\right]$$

Since we assume $\widetilde{\gamma}(\boldsymbol{p})=1$, $\nu_j=-1$, and using property 3 and the fact that $\boldsymbol{p}\in\mathcal{P}$, we get that:

$$-\widetilde{\gamma}(\boldsymbol{p})v_l\nu_j\langle\frac{\partial L_{\Psi(\mathcal{D})}}{\partial\boldsymbol{w}_l^{(t)}},\boldsymbol{p}\rangle = v_l\langle\frac{\partial L_{\Psi(\mathcal{D})}}{\partial\boldsymbol{w}_l^{(t)}},\boldsymbol{p}\rangle$$

$$= \frac{4}{n_i}(\lambda-\mathbb{E}\left[y\right])\mathbb{P}_{(\boldsymbol{x},y)\sim\mathcal{D}^{(i)}}\left[(x_{2j-1},x_{2j})=\widetilde{\boldsymbol{p}}\right] > \frac{\Delta\epsilon}{n_i}$$

Using the fact that $\lambda=\mathbb{E}\left[y\right]+\frac{\Delta}{4}$.

- Otherwise, observe that:

$$\langle\frac{\partial L_{\Psi(\mathcal{D})}}{\partial\boldsymbol{w}_l^{(t)}},\boldsymbol{p}\rangle = \frac{4v_l q_{\widetilde{\boldsymbol{p}}}}{n_i}\mathbb{E}_{(\boldsymbol{z},y)\sim\mathcal{D}^{(i-1)}}\left[(\lambda-y)\mathbf{1}\{z_j=\widetilde{\gamma}(\boldsymbol{p})\}\right]$$

$$= \frac{4v_l q_{\widetilde{\boldsymbol{p}}}}{n_i}\left(\lambda\mathbb{P}_{(\boldsymbol{z},y)\sim\mathcal{D}^{(i-1)}}\left[z_j=\widetilde{\gamma}(\boldsymbol{p})\right]-\mathbb{E}_{(\boldsymbol{z},y)\sim\mathcal{D}^{(i-1)}}\left[y\frac{1}{2}(z_j\cdot\widetilde{\gamma}(\boldsymbol{p})+1)\right]\right)$$

$$= \frac{2v_l q_{\widetilde{\boldsymbol{p}}}}{n_i}\left(2\lambda\mathbb{P}_{(\boldsymbol{z},y)\sim\mathcal{D}^{(i-1)}}\left[z_j=\widetilde{\gamma}(\boldsymbol{p})\right]-\widetilde{\gamma}(\boldsymbol{p})c_{i-1,j}-\mathbb{E}_{(\boldsymbol{z},y)\sim\mathcal{D}^{(i-1)}}\left[y\right]\right)$$

And therefore we get:

$$-\widetilde{\gamma}(\boldsymbol{p})v_l\,\mathrm{sign}(c_{i-1,j})\langle\frac{\partial L_{\Psi(\mathcal{D})}}{\partial\boldsymbol{w}_l^{(t)}},\boldsymbol{p}\rangle = \frac{2q_{\widetilde{\boldsymbol{p}}}}{n_i}\left(|c_{i-1,j}|+\mathrm{sign}(c_{i-1,j})\widetilde{\gamma}(\boldsymbol{p})(\mathbb{E}\left[y\right]-2\lambda\mathbb{P}\left[z_j=\widetilde{\gamma}(\boldsymbol{p})\right])\right)$$

Now, if $\operatorname{sign}(c_{i-1,j})\widetilde{\gamma}(\boldsymbol{p}) = 1$, using property 1, since $\mathcal{I}_{i-1,j} \neq 0$ we get:

$$-\widetilde{\gamma}(\boldsymbol{p})v_l \operatorname{sign}(c_{i-1,j})\langle\frac{\partial L_{\Psi(\mathcal{D})}}{\partial \boldsymbol{w}_l^{(t)}}, \boldsymbol{p}\rangle \geq \frac{q_{\widetilde{\boldsymbol{p}}}}{n_i}\left(|c_{i-1,j}| + \mathbb{E}[y] - 2\lambda\right) > \frac{\epsilon}{n_i}\Delta$$

Otherwise, we have $\operatorname{sign}(c_{i-1,j})\widetilde{\gamma}(\boldsymbol{p}) = -1$, and then:

$$-\widetilde{\gamma}(\boldsymbol{p})v_l \operatorname{sign}(c_{i-1,j})\langle\frac{\partial L_{\Psi(\mathcal{D})}}{\partial \boldsymbol{w}_l^{(t)}}, \boldsymbol{p}\rangle \geq \frac{q_{\widetilde{\boldsymbol{p}}}}{n_i}\left(|c_{i-1,j}| - \mathbb{E}[y]\right) > \frac{2\epsilon}{n_i}\Delta$$

where we use property 3 and the fact that $\boldsymbol{p} \in \mathcal{P}$.

$\square$

We introduce the following notation: for a sample $S \subseteq \mathcal{X}' \times \mathcal{Y}$, and some function $f : \mathcal{X}' \to \mathcal{X}'$, denote by $f(S)$ the sample $f(S) := \{(f(\boldsymbol{x}), y)\}_{(\boldsymbol{x},y)\in S}$. Using standard concentration of measure arguments, we get that the gradient on the sample approximates the gradient on the distribution:

**Lemma 4.** *Fix $\delta > 0$. Assume we sample $S \sim \mathcal{D}$, with $|S| > \frac{2^{11}}{\epsilon^2 \Delta^2}\log\frac{8}{\delta}$. Then, with probability at least $1 - \delta$, for every $\boldsymbol{p} \in \{\pm 1\}^2$ such that $\langle\boldsymbol{w}_l^{(t)}, \boldsymbol{p}\rangle > 0$ it holds that:*

$$\left|\langle\frac{\partial L_{\Psi(\mathcal{D})}}{\partial \boldsymbol{w}_l^{(t)}}, \boldsymbol{p}\rangle - \langle\frac{\partial L_{\Psi(S)}}{\partial \boldsymbol{w}_l^{(t)}}, \boldsymbol{p}\rangle\right| \leq \frac{\epsilon}{4n_i}\Delta$$

*Proof.* Fix some $\boldsymbol{p} \in \{\pm 1\}^2$ with $\langle\boldsymbol{w}_l^{(t)}, \boldsymbol{p}\rangle > 0$. Similar to what we previously showed, we get that:

$$\langle\frac{\partial L_{\Psi(S)}}{\partial \boldsymbol{w}_l^{(t)}}, \boldsymbol{p}\rangle$$
$$= \frac{2}{n_i}\mathbb{E}_{(\boldsymbol{x},y)\sim\Psi(S)}\left[(\lambda - y)v_l\mathbf{1}\{g_t(x_{2j-1}, x_{2j}) \in (-1,1)\} \cdot \mathbf{1}\{\langle\boldsymbol{w}_l^{(t)}, (x_{2j-1}, x_{2j})\rangle \geq 0\} \cdot \langle(x_{2j-1}, x_{2j}), \boldsymbol{p}\rangle\right]$$
$$= \frac{2}{n_i}\mathbb{E}_{(\boldsymbol{x},y)\sim\Psi(S)}\left[(\lambda - y)v_l\mathbf{1}\{g_t(x_{2j-1}, x_{2j}) \in (-1,1)\} \cdot \mathbf{1}\{(x_{2j-1}, x_{2j}) = \boldsymbol{p}\}\|\boldsymbol{p}\|^2\right]$$
$$= \frac{4}{n_i}\mathbb{E}_{(\boldsymbol{x},y)\sim\Psi(S)}\left[(\lambda - y)v_l\mathbf{1}\{g_t(x_{2j-1}, x_{2j}) \in (-1,1)\} \cdot \mathbf{1}\{(x_{2j-1}, x_{2j}) = \boldsymbol{p}\}\right]$$

Denote $f(\boldsymbol{x}, y) = (\lambda - y)v_l\mathbf{1}\{g_t(x_{2j-1}, x_{2j}) \in (-1,1)\} \cdot \mathbf{1}\{(x_{2j-1}, x_{2j}) = \boldsymbol{p}\}$, and notice that since $\lambda \leq 1$, we have $f(\boldsymbol{x}, y) \in [-2, 2]$. Now, from Hoeffding's inequality we get that:

$$\mathbb{P}_S\left[\left|\mathbb{E}_{\Psi(S)}[f(\boldsymbol{x}, y)] - \mathbb{E}_{\Psi(\mathcal{D})}[f(\boldsymbol{x}, y)]\right| \geq \tau\right] \leq 2\exp\left(-\frac{1}{8}|S|\tau^2\right)$$

So, for $|S| > \frac{8}{\tau^2}\log\frac{8}{\delta}$ we get that with probability at least $1 - \frac{\delta}{4}$ we have:

$$\left|\langle\frac{\partial L_{\Psi(\mathcal{D})}}{\partial \boldsymbol{w}_l^{(t)}}, \boldsymbol{p}\rangle - \langle\frac{\partial L_{\Psi(S)}}{\partial \boldsymbol{w}_l^{(t)}}, \boldsymbol{p}\rangle\right| = \frac{4}{n_i}\left|\mathbb{E}_{\Psi(S)}[f(\boldsymbol{x}, y)] - \mathbb{E}_{\Psi(\mathcal{D})}[f(\boldsymbol{x}, y)]\right| < \frac{4}{n_i}\tau$$

Taking $\tau = \frac{\epsilon}{16}\Delta$ and using the union bound over all $\boldsymbol{p} \in \{\pm 1\}^2$ completes the proof. $\square$

Using the two previous lemmas, we can estimate the behavior of the gradient on the sample, with respect to a given pattern $\boldsymbol{p}$:

**Lemma 5.** *Fix $\delta > 0$. Assume we sample $S \sim \mathcal{D}$, with $|S| > \frac{2^{11}}{\epsilon^2 \Delta^2}\log\frac{8}{\delta}$. Then, with probability at least $1 - \delta$, for every $\boldsymbol{p} \in \mathcal{P}$, and for every $l \in [k]$ such that $\langle\boldsymbol{w}_l^{(t)}, \boldsymbol{p}\rangle > 0$ and $g_t(\boldsymbol{p}) \in (-1,1)$, the following holds:*

$$-\widetilde{\gamma}(\boldsymbol{p})v_l\nu_j\langle\frac{\partial L_{\Psi(S)}}{\partial \boldsymbol{w}_l^{(t)}}, \boldsymbol{p}\rangle > \frac{\epsilon}{2n_i}\Delta$$

*Proof.* Using Lemma 3 and Lemma 4, with probability at least $1 - \delta$:

$$-\widetilde{\gamma}(\boldsymbol{p})v_l\nu_j\langle\frac{\partial L_{\Psi(S)}}{\partial \boldsymbol{w}_l^{(t)}},\boldsymbol{p}\rangle = -\widetilde{\gamma}(\boldsymbol{p})v_l\nu_j\left(\langle\frac{\partial L_{\Psi(\mathcal{D})}}{\partial \boldsymbol{w}_l^{(t)}},\boldsymbol{p}\rangle + \langle\frac{\partial L_{\Psi(S)}}{\partial \boldsymbol{w}_l^{(t)}},\boldsymbol{p}\rangle - \langle\frac{\partial L_{\Psi(\mathcal{D})}}{\partial \boldsymbol{w}_l^{(t)}},\boldsymbol{p}\rangle\right)$$

$$\geq -\widetilde{\gamma}(\boldsymbol{p})v_l\nu_j\langle\frac{\partial L_{\Psi(\mathcal{D})}}{\partial \boldsymbol{w}_l^{(t)}},\boldsymbol{p}\rangle - \left|\langle\frac{\partial L_{\Psi(S)}}{\partial \boldsymbol{w}_l^{(t)}},\boldsymbol{p}\rangle - \langle\frac{\partial L_{\Psi(\mathcal{D})}}{\partial \boldsymbol{w}_l^{(t)}},\boldsymbol{p}\rangle\right|$$

$$> \frac{\epsilon}{n_i}\Delta - \frac{\epsilon}{4n_i}\Delta \geq \frac{3\epsilon}{4n_i}\Delta$$

$\square$

We want to show that if the value of $g_t$ gets "stuck", then it recovered the value of the gate, multiplied by the correlation $c_{i-1,j}$. We do this by observing the dynamics of $\langle\boldsymbol{w}_l^{(t)},\boldsymbol{p}\rangle$. In most cases, its value moves in the right direction, except for a small set that oscillates around zero. This set is the following:

$$A_t = \left\{(l,\boldsymbol{p}) \ : \ \boldsymbol{p} \in \mathcal{P} \wedge \widetilde{\gamma}(\boldsymbol{p})v_l\nu_j < 0 \wedge \langle\boldsymbol{w}_l^{(t)},\boldsymbol{p}\rangle \leq \frac{8\eta}{n_i} \wedge (\widetilde{\gamma}(-\boldsymbol{p})v_l\nu_j < 0 \vee -\boldsymbol{p} \in \mathcal{P})\right\}$$

We have the following simple observation:

**Lemma 6.** *With the assumptions of Lemma 5, with probability at least $1 - \delta$, for every $t$ we have:*
$A_t \subseteq A_{t+1}$.

*Proof.* Fix some $(l,\boldsymbol{p}) \in A_t$, and we need to show that $\langle\boldsymbol{w}_l^{(t+1)},\boldsymbol{p}\rangle \leq \frac{8\eta}{n_i}$. If $\langle\boldsymbol{w}_l^{(t)},\boldsymbol{p}\rangle = 0$ then[4] $\langle\boldsymbol{w}_l^{(t+1)},\boldsymbol{p}\rangle = \langle\boldsymbol{w}_l^{(t)},\boldsymbol{p}\rangle \leq \frac{8\eta}{n_i}$ and we are done. If $\langle\boldsymbol{w}_l^{(t)},\boldsymbol{p}\rangle > 0$ then, since $\boldsymbol{p} \in \mathcal{P}$ we have from Lemma 5, w.p at least $1 - \delta$:

$$-\langle\frac{\partial L_{\Psi(S)}}{\partial \boldsymbol{w}_l^{(t)}},\boldsymbol{p}\rangle < \widetilde{\gamma}(\boldsymbol{p})v_l\nu_j\frac{\epsilon}{2n_i}\Delta < 0$$

Where we use the fact that $\widetilde{\gamma}(\boldsymbol{p})v_l\nu_j < 0$. Therefore, we get:

$$\langle\boldsymbol{w}_l^{(t+1)},\boldsymbol{p}\rangle = \langle\boldsymbol{w}_l^{(t)},\boldsymbol{p}\rangle - \eta\langle\frac{\partial L_{\Psi(S)}}{\partial \boldsymbol{w}_l^{(t)}},\boldsymbol{p}\rangle \leq \langle\boldsymbol{w}_l^{(t)},\boldsymbol{p}\rangle \leq \frac{8\eta}{n_i}$$

Otherwise, we have $\langle\boldsymbol{w}_l^{(t)},\boldsymbol{p}\rangle < 0$, so:

$$\langle\boldsymbol{w}_l^{(t+1)},\boldsymbol{p}\rangle = \langle\boldsymbol{w}_l^{(t)},\boldsymbol{p}\rangle - \eta\langle\frac{\partial L_{\Psi(S)}}{\partial \boldsymbol{w}_l^{(t)}},\boldsymbol{p}\rangle \leq \langle\boldsymbol{w}_l^{(t)},\boldsymbol{p}\rangle + \frac{8\eta}{n_i} \leq \frac{8\eta}{n_i}$$

$\square$

Now, we want to show that all $\langle\boldsymbol{w}_l^{(t)},\boldsymbol{p}\rangle$ with $(l,\boldsymbol{p}) \notin A_t$ and $\boldsymbol{p} \in \mathcal{P}$ move in the direction of $\widetilde{\gamma}(\boldsymbol{p})\cdot\nu_j$:

**Lemma 7.** *With the assumptions of Lemma 5, with probability at least $1-\delta$, for every $l,t$ and $\boldsymbol{p} \in \mathcal{P}$ such that $\langle\boldsymbol{w}_l^{(t)},\boldsymbol{p}\rangle > 0$ and $(l,\boldsymbol{p}) \notin A_t$, it holds that:*

$$\left(\sigma(\langle\boldsymbol{w}_l^{(t)},\boldsymbol{p}\rangle) - \sigma(\langle\boldsymbol{w}_l^{(t-1)},\boldsymbol{p}\rangle)\right) \cdot \widetilde{\gamma}(\boldsymbol{p})v_l\nu_j \geq 0$$

*Proof.* Assume the result of Lemma 5 holds (this happens with probability at least $1-\delta$). We cannot have $\langle\boldsymbol{w}_l^{(t-1)},\boldsymbol{p}\rangle = 0$, since otherwise we would have $\langle\boldsymbol{w}_l^{(t)},\boldsymbol{p}\rangle = 0$, contradicting the assumption. If $\langle\boldsymbol{w}_l^{(t-1)},\boldsymbol{p}\rangle > 0$, since we require $\langle\boldsymbol{w}_l^{(t)},\boldsymbol{p}\rangle > 0$ we get that:

$$\sigma(\langle\boldsymbol{w}_l^{(t)},\boldsymbol{p}\rangle) - \sigma(\langle\boldsymbol{w}_l^{(t-1)},\boldsymbol{p}\rangle) = \langle\boldsymbol{w}_l^{(t)} - \boldsymbol{w}_l^{(t-1)},\boldsymbol{p}\rangle = -\eta\langle\frac{\partial L_{\Psi(S)}}{\partial \boldsymbol{w}_l^{(t-1)}},\boldsymbol{p}\rangle$$

and the required follows from Lemma 5. Otherwise, we have $\langle\boldsymbol{w}_l^{(t-1)},\boldsymbol{p}\rangle < 0$. We observe the following cases:

- If $\widetilde{\gamma}(\boldsymbol{p})v_l\nu_j \geq 0$ then we are done, since:

$$\left(\sigma(\langle \boldsymbol{w}_l^{(t)}, \boldsymbol{p}\rangle) - \sigma(\langle \boldsymbol{w}_l^{(t-1)}, \boldsymbol{p}\rangle)\right) \cdot \widetilde{\gamma}(\boldsymbol{p})v_l\nu_j = \sigma(\langle \boldsymbol{w}_l^{(t)}, \boldsymbol{p}\rangle) \cdot \widetilde{\gamma}(\boldsymbol{p})v_l\nu_j \geq 0$$

- Otherwise, we have $\widetilde{\gamma}(\boldsymbol{p})v_l\nu_j < 0$. We also have:

$$\langle \boldsymbol{w}_l^{(t)}, \boldsymbol{p}\rangle = \langle \boldsymbol{w}_l^{(t-1)}, \boldsymbol{p}\rangle - \eta \langle \frac{\partial L_{\Psi(S)}}{\partial \boldsymbol{w}_l^{(t)}}, \boldsymbol{p}\rangle \leq \langle \boldsymbol{w}_l^{(t-1)}, \boldsymbol{p}\rangle + \frac{8\eta}{n_i} \leq \frac{8\eta}{n_i}$$

Since we assume $(l, \boldsymbol{p}) \notin A_t$, we must have $-\boldsymbol{p} \in \mathcal{P}$ and $\widetilde{\gamma}(-\boldsymbol{p})v_l\nu_j \geq 0$. Therefore, from Lemma 5 we get:

$$\langle \frac{\partial L_{\Psi(S)}}{\partial \boldsymbol{w}_l^{(t)}}, -\boldsymbol{p}\rangle < -\widetilde{\gamma}(-\boldsymbol{p})v_l\nu_j \frac{\epsilon}{2n_i}\Delta$$

And hence:

$$0 < \langle \boldsymbol{w}_l^{(t)}, \boldsymbol{p}\rangle = \langle \boldsymbol{w}_l^{(t-1)}, \boldsymbol{p}\rangle + \eta \langle \frac{\partial L_{\Psi(S)}}{\partial \boldsymbol{w}_l^{(t-1)}}, -\boldsymbol{p}\rangle \leq -\eta\widetilde{\gamma}(-\boldsymbol{p})v_l\nu_j \frac{\epsilon}{2n_i}\Delta < 0$$

and we reach a contradiction.

$\qquad\square$

From the above, we get the following:

**Corollary 1.** *With the assumptions of Lemma 5, with probability at least $1 - \delta$, for every $l, t$ and $\boldsymbol{p} \in \mathcal{P}$ such that $\langle \boldsymbol{w}_l^{(t)}, \boldsymbol{p}\rangle > 0$ and $(l, \boldsymbol{p}) \notin A_t$, the following holds:*

$$\left(\sigma(\langle \boldsymbol{w}_l^{(t)}, \boldsymbol{p}\rangle) - \sigma(\langle \boldsymbol{w}_l^{(0)}, \boldsymbol{p}\rangle)\right) \cdot \widetilde{\gamma}(\boldsymbol{p})v_l\nu_j \geq 0$$

*Proof.* Notice that for every $t' \leq t$ we have $(l, \boldsymbol{p}) \notin A_{t'} \subseteq A_t$. Therefore, using the previous lemma:

$$\left(\sigma(\langle \boldsymbol{w}_l^{(t)}, \boldsymbol{p}\rangle) - \sigma(\langle \boldsymbol{w}_l^{(0)}, \boldsymbol{p}\rangle)\right) \cdot \widetilde{\gamma}(\boldsymbol{p})v_l\nu_j = \sum_{1 \leq t' \leq t} \left(\sigma(\langle \boldsymbol{w}_l^{(t)}, \boldsymbol{p}\rangle) - \sigma(\langle \boldsymbol{w}_l^{(t')}, \boldsymbol{p}\rangle)\right) \cdot \widetilde{\gamma}(\boldsymbol{p})v_l\nu_j \geq 0$$

$\qquad\square$

Finally, we need to show that there are some "good" neurons, that are moving strictly away from zero:

**Lemma 8.** *Fix $\delta > 0$. Assume we sample $S \sim \mathcal{D}$, with $|S| > \frac{2^{11}}{\epsilon^2\Delta^2}\log\frac{8}{\delta}$. Assume that $k \geq \log^{-1}(\frac{4}{3})\log(\frac{4}{\delta})$. Then with probability at least $1 - 2\delta$, for every $\boldsymbol{p} \in \mathcal{P}$, there exists $l \in [k]$ such that for every $t$ with $g_{t-1}(\boldsymbol{p}) \in (-1, 1)$, we have:*

$$\sigma(\langle \boldsymbol{w}_l^{(t)}, \boldsymbol{p}\rangle) \cdot \widetilde{\gamma}(\boldsymbol{p})v_l\nu_j \geq \eta t \frac{\epsilon}{2n_i}\Delta$$

*Proof.* Assume the result of Lemma 5 holds (happens with probability at least $1 - \delta$). Fix some $\boldsymbol{p} \in \mathcal{P}$. For $l \in [k]$, with probability $\frac{1}{4}$ we have both $v_l = \widetilde{\gamma}(\boldsymbol{p})\nu_j$ and $\langle \boldsymbol{w}_l^{(0)}, \boldsymbol{p}\rangle > 0$. Therefore, the probability that there exists $l \in [k]$ such that the above holds is $1 - (\frac{3}{4})^k \geq 1 - \frac{\delta}{4}$. Using the union bound, w.p at least $1 - \delta$, there exists such $l \in [k]$ for every $\boldsymbol{p} \in \{\pm 1\}^2$. In such case, we have $\langle \boldsymbol{w}_l^{(t)}, \boldsymbol{p}\rangle \geq \eta t \frac{\epsilon}{2n_i}\Delta$, by induction:

- For $t = 0$ this is true since $\langle \boldsymbol{w}_l^{(0)}, \boldsymbol{p}\rangle > 0$.

- If the above holds for $t-1$, then $\langle w_l^{(t-1)}, p \rangle > 0$, and therefore, using $v_l = \widetilde{\gamma}(p)\nu_j$ and Lemma 5:

$$-\langle \frac{\partial L_{\Psi(\mathcal{D})}}{\partial w_l^{(t)}}, p \rangle > \widetilde{\gamma}(p)v_l\nu_j \frac{\epsilon}{2n_i}\Delta$$

And we get:

$$\begin{aligned}
\langle w_l^{(t)}, p \rangle &= \langle w_l^{(t-1)}, p \rangle - \eta\langle \frac{\partial L_{\Psi(\mathcal{D})}}{\partial w_l^{(t)}}, p \rangle \\
&> \langle w_l^{(t-1)}, p \rangle + \eta\widetilde{\gamma}(p)v_l\nu_j \frac{\epsilon}{2n_i}\Delta \\
&\geq \eta(t-1)\frac{\epsilon}{2n_i}\Delta + \eta\frac{\epsilon}{2n_i}\Delta
\end{aligned}$$

$\square$

Using the above results, we can analyze the behavior of $g_t(p)$:

**Lemma 9.** *Assume we initialize $w_l^{(0)}$ such that $\left\| w_l^{(0)} \right\| \leq \frac{1}{4k}$. Fix $\delta > 0$. Assume we sample $S \sim \mathcal{D}$, with $|S| > \frac{2^{11}}{\epsilon^2\Delta^2}\log\frac{8}{\delta}$. Then with probability at least $1 - 2\delta$, for every $p \in \mathcal{P}$, for $t > \frac{6n_i}{\sqrt{2}\eta\epsilon\Delta}$ we have:*

$$g_t(p) = \widetilde{\gamma}(p)\nu_j$$

*Proof.* Using Lemma 8, w.p at least $1 - 2\delta$, for every such $p$ there exists $l_p \in [k]$ such that for every $t$ with $g_{t-1}(p) \in (-1, 1)$:

$$v_{l_p}\sigma(\langle w_{l_p}^{(t)}, p \rangle) \cdot \widetilde{\gamma}(p)\nu_j \geq \eta t\frac{\epsilon}{2n_i}\Delta$$

Assume this holds, and fix some $p \in \mathcal{P}$. Let $t$, such that $g_{t-1}(p) \in (-1, 1)$. Denote the set of indexes $J = \{l : \langle w_l^{(t)}, p \rangle > 0\}$. We have the following:

$$\begin{aligned}
g_t(p) &= \sum_{l \in J} v_l\sigma(\langle w_l^{(t)}, p \rangle) \\
&= v_{l_p}\sigma(\langle w_{l_p}^{(t)}, p \rangle) + \sum_{l \in J\backslash\{l_p\},(l,p)\notin A_t} v_l\sigma(\langle w_l^{(t)}, p \rangle) + \sum_{l \in J\backslash\{l_p\},(l,p)\in A_t} v_l\sigma(\langle w_l^{(t)}, p \rangle)
\end{aligned}$$

From Corollary 1 we have:

$$\widetilde{\gamma}(p)\nu_j \cdot \sum_{l \in J\backslash\{l_p\},(l,p)\notin A_t} v_l\sigma(\langle w_l^{(t)}, p \rangle) \geq -k\sigma(\langle w_l^{(0)}, p \rangle) \geq -\frac{1}{4}$$

By definition of $A_t$ and by our assumption on $\eta$ we have:

$$\widetilde{\gamma}(p)\nu_j \cdot \sum_{l \in J\backslash\{l_p\},(l,p)\in A_t} v_l\sigma(\langle w_l^{(t)}, p \rangle) \geq -k\frac{8\eta}{n_i} \geq -\frac{1}{4}$$

Therefore, we get:

$$\widetilde{\gamma}(p)\nu_j \cdot g_t(p) \geq \eta t\frac{\epsilon}{2\sqrt{2}n_i}\Delta - \frac{1}{2}$$

This shows that for $t > \frac{6n_i}{\sqrt{2}\eta\epsilon\Delta}$ we get the required. $\square$

*Proof.* of Lemma 2. Using the result of Lemma 9, with union bound over all choices of $j \in [n_i/2]$. The required follows by the definition of $\widetilde{\gamma}(x_{2j-1}, x_{2j}) = \gamma_{i-1,j}(\xi_{2j-1}x_{2j-1}, \xi_{2j}x_{2j})$. $\square$

# B  Proofs of Section 3.2

*Proof.* of Lemma 1. Property 1 is immediate from assumption 1. For property 2, fix some $i \in [d], j \in [n_i/2], \boldsymbol{p} \in \{\pm 1\}^2, y' \in \{\pm 1\}$, such that:

$$\mathbb{P}_{(\boldsymbol{x},y) \sim \mathcal{D}^{(i)}} \left[ \gamma_{i-1,j}(x_{2j-1}, x_{2j}) = \gamma_{i-1,j}(\boldsymbol{p}) \right] > 0$$

Assume w.l.o.g. that $j = 1$. Denote by $W$ the set of all possible choices for $x_3, \ldots, x_{n_i}$, such that when $(x_1, x_2) = \boldsymbol{p}$, the resulting label is $y'$. Formally:

$$W := \{ (x_3, \ldots, x_{n_i}) \ : \ \Gamma_{i \ldots d}(p_1, p_2, x_3, \ldots, x_{n_i}) = y' \}$$

Then we get:

$$\mathbb{P}_{\mathcal{D}^{(i)}} \left[ (x_1, x_2) = \boldsymbol{p}, y = y', \gamma_{i-1,j}(x_1, x_2) = \gamma_{i-1,j}(\boldsymbol{p}) \right]$$
$$= \mathbb{P}_{\mathcal{D}^{(i)}} \left[ (x_1, x_2) = \boldsymbol{p}, (x_3, \ldots, x_{n_i}) \in W, \gamma_{i-1,j}(x_1, x_2) = \gamma_{i-1,j}(\boldsymbol{p}) \right]$$
$$= \mathbb{P}_{\mathcal{D}^{(i)}} \left[ (x_1, x_2) = \boldsymbol{p}, \gamma_{i-1,j}(x_1, x_2) = \gamma_{i-1,j}(\boldsymbol{p}) \right] \cdot \mathbb{P}_{\mathcal{D}^{(i)}} \left[ (x_3, \ldots, x_{n_i}) \in W \right]$$
$$= \mathbb{P}_{\mathcal{D}^{(i)}} \left[ (x_1, x_2) = \boldsymbol{p} | \gamma_{i-1,j}(x_1, x_2) = \gamma_{i-1,j}(\boldsymbol{p}) \right] \cdot \mathbb{P}_{\mathcal{D}^{(i)}} \left[ \gamma_{i-1,j}(x_1, x_2) = \gamma_{i-1,j}(\boldsymbol{p}), (x_3, \ldots, x_{n_i}) \in W \right]$$
$$= \mathbb{P}_{\mathcal{D}^{(i)}} \left[ (x_1, x_2) = \boldsymbol{p} | \gamma_{i-1,j}(x_1, x_2) = \gamma_{i-1,j}(\boldsymbol{p}) \right] \cdot \mathbb{P}_{\mathcal{D}^{(i)}} \left[ y = y', \gamma_{i-1,j}(x_1, x_2) = \gamma_{i-1,j}(\boldsymbol{p}) \right]$$

And dividing by $\mathbb{P}_{\mathcal{D}^{(i)}} \left[ \gamma_{i-1,j}(x_1, x_2) = \gamma_{i-1,j}(\boldsymbol{p}) \right]$ gives the required.

For property 3, we observe two cases. If $c_{i,j} \geq 0$ then:

$$\Delta \leq c_{i,j} - \mathbb{E}\left[y\right] = \mathbb{E}\left[x_j y - y\right] = \mathbb{E}\left[y(x_j - 1)\right]$$
$$= 2\mathbb{P}\left[x_j = -1 \wedge y = -1\right] - 2\mathbb{P}\left[x_j = -1 \wedge y = 1\right]$$
$$\leq 2\mathbb{P}\left[x_j = -1 \wedge y = -1\right] \leq 2\mathbb{P}\left[x_j = -1\right]$$

Otherwise, if $c_{i,j} < 0$ we have:

$$\Delta \leq -c_{i,j} - \mathbb{E}\left[y\right] = \mathbb{E}\left[-x_j y - y\right] = -\mathbb{E}\left[y(x_j + 1)\right]$$
$$= 2\mathbb{P}\left[x_j = 1 \wedge y = -1\right] - 2\mathbb{P}\left[x_j = 1 \wedge y = 1\right]$$
$$\leq 2\mathbb{P}\left[x_j = 1 \wedge y = -1\right] \leq 2\mathbb{P}\left[x_j = 1\right]$$

So, in any case $\mathbb{P}\left[x_j = 1\right] \in (\frac{\Delta}{2}, 1 - \frac{\Delta}{2})$, and since every bit in every layer is independent, we get property 3 holds with $\epsilon = \frac{\Delta^2}{4}$. □

# C  Proofs of Section 3.3

## C.1  Parity Circuits

We observe the $k$-parity problem, where the target function is $f(\boldsymbol{x}) = \prod_{j \in I} x_j$ some subset $I \subseteq [n]$ of size $|I| = k$. A simple construction shows that $f$ can be implemented by a tree structured circuit as defined previously. We define the gates of the first layer by:

$$\gamma_{d-1,j}(z_1, z_2) = \begin{cases} z_1 z_2 & x_{2j-1}, x_{2j} \in I \\ z_1 & x_{2j-1} \in I, x_{2j} \notin I \\ z_2 & x_{2j} \in I, x_{2j-1} \notin I \\ 1 & o.w \end{cases}$$

And for all other layers $i < d - 1$, we define: $\gamma_{i,j}(z_1, z_1) = z_1 z_2$. Then we get the following:

**Lemma 10.** *Let $C$ be a Boolean circuit as defined above. Then: $h_C(\boldsymbol{x}) = \prod_{j \in I} x_j = f(\boldsymbol{x})$.*

Now, let $\mathcal{D}_{\mathcal{X}}$ be some product distribution over $\mathcal{X}$, and denote $p_j := \mathbb{P}_{\mathcal{D}_{\mathcal{X}}}\left[x_j = 1\right]$. Let $\mathcal{D}$ be the distribution of $(\boldsymbol{x}, f(\boldsymbol{x}))$ where $\boldsymbol{x} \sim \mathcal{D}_{\mathcal{X}}$. Then for the circuit defined above we get the following:

**Lemma 11.** *Fix some $\xi \in (0, \frac{1}{4})$. For every product distribution $\mathcal{D}$ with $p_j \in (\xi, \frac{1}{2} - \xi) \cup (\frac{1}{2} + \xi, 1 - \xi)$ for all $j$, if $\mathcal{I}_{i,j} \neq 0$ then $|c_{i,j}| - |\mathbb{E}\left[y\right]| \geq (2\xi)^k$ and $\mathbb{P}_{(\boldsymbol{z},y) \sim \Gamma_{(i+1) \ldots d}(\mathcal{D})}\left[z_j = 1\right] \in (\xi, 1 - \xi)$.*

The above lemma shows that every non-degenerate product distribution that is far enough from the uniform distribution, satisfies assumption 1 with $\Delta = (2\xi)^k$. Using the fact that at each layer, the output of each gate is an independent random variable (since the input distribution is a product distribution), we get that property 3 is satisfied with $\epsilon = \xi^2$. This gives us the following result:

**Corollary 2.** *Let $\mathcal{D}$ be a product distribution with $p_j \in (\xi, \frac{1}{2} - \xi) \cup (\frac{1}{2} + \xi, 1 - \xi)$ for every $j$, with the target function being a ($\log n$)-parity (i.e., $k = \log n$). Then, when running algorithm 1 as described in Theorem 2, with probability at least $1 - \delta$ the algorithm returns the true target function $h_C$, with run-time and sample complexity polynomial in $n$.*

*Proof.* of Lemma 10.

For every gate $(i, j)$, let $J_{i,j}$ be the subset of leaves in the binary tree whose root is the node $(i, j)$. Namely, $J_{i,j} := \{(j-1)2^{d-i} + 1, \ldots, j2^{d-i}\}$. Then we show inductively that for an input $x \in \{\pm 1\}^n$, the $(i, j)$ gate outputs: $\prod_{l \in I \cap J_{i,j}} x_l$:

- For $i = d - 1$, this is immediate from the definition of the gate $\gamma_{d-1,j}$.

- Assume the above is true for some $i$ and we will show this for $i - 1$. By definition of the circuit, the output of the $(i - 1, j)$ gate is a product of the output of its inputs from the previous layers, the gates $(i, 2j - 1)$, $(i, 2j)$. By the inductive assumption, we get that the output of the $(i - 1, j)$ gate is therefore:

$$\left( \prod_{l \in J_{i,2j-1} \cap I} x_l \right) \cdot \left( \prod_{l \in J_{i,2j} \cap I} x_l \right) = \prod_{l \in (J_{i,j2-1} \cup J_{i,2j}) \cap I} x_l = \prod_{l \in J_{i-1,j}} x_l$$

From the above, the output of the target circuit is $\prod_{l \in J_{0,1} \cap I} x_l = \prod_{l \in I} x_l$, as required. $\square$

*Proof.* of Lemma 11.

By definition we have:

$c_{i,j} = \mathbb{E}_{(x,y) \sim \mathcal{D}} \left[ \Gamma_{(i+1)\ldots d}(x)_j y \right] = \mathbb{E}_{(x,y) \sim \mathcal{D}} \left[ \Gamma_{(i+1)\ldots d}(x)_j y \right] = \mathbb{E}_{(x,y) \sim \mathcal{D}} \left[ \Gamma_{(i+1)\ldots d}(x)_j x_1 \cdots x_k \right]$

Since we require $\mathcal{I}_{i,j} \neq 0$, then we cannot have $\Gamma_{(i+1)\ldots d}(x)_j \equiv 1$. So, from what we showed previously, it follows that $\Gamma_{(i+1)\ldots d}(x)_j = \prod_{j' \in I'} x_{j'}$ for some $\emptyset \neq I' \subseteq I$. Therefore, we get that:

$$c_{i,j} = \mathbb{E}_{\mathcal{D}} \left[ \prod_{j' \in I \setminus I'} x_{j'} \right] = \prod_{j' \in I \setminus I'} \mathbb{E}_{\mathcal{D}} \left[ x_{j'} \right] = \prod_{j' \in I \setminus I'} (2p_{j'} - 1)$$

Furthermore, we have that:

$$\mathbb{E}_{\mathcal{D}} \left[ y \right] = \mathbb{E}_{\mathcal{D}} \left[ \prod_{j' \in I} x_{j'} \right] = \prod_{j' \in I} \mathbb{E}_{\mathcal{D}} \left[ x_{j'} \right] = \prod_{j' \in I} (2p_{j'} - 1)$$

And using the assumption on $p_j$ we get:

$$|c_{i,j}| - |\mathbb{E}_{\mathcal{D}} \left[ y \right]| = \prod_{j' \in [k] \setminus I'} |2p_{j'} - 1| - \prod_{j' \in [k]} |2p_{j'} - 1|$$

$$= \left( \prod_{j' \in [k] \setminus I'} |2p_{j'} - 1| \right) \left( 1 - \prod_{j' \in I'} |2p_{j'} - 1| \right)$$

$$\geq \left( \prod_{j' \in [k] \setminus I'} |2p_{j'} - 1| \right) \left( 1 - (1 - 2\xi)^{|I'|} \right)$$

$$\geq (2\xi)^{k - |I'|} \left( 1 - (1 - 2\xi) \right) \geq (2\xi)^k$$

Now, for the second result, we have:

$$\mathbb{P}_{(\boldsymbol{z},y)\sim\Gamma_{i\ldots d}(\mathcal{D})}\left[z_j = 1\right] = \mathbb{E}_{(\boldsymbol{x},y)\sim\mathcal{D}}\left[\mathbf{1}\{\Gamma_{(i+1)\ldots d}(\boldsymbol{x})_j = 1\}\right]$$

$$= \mathbb{E}_{(\boldsymbol{x},y)\sim\mathcal{D}}\left[\frac{1}{2}(\prod_{j'\in I'} x_{j'} + 1)\right]$$

$$= \frac{1}{2}\prod_{j'\in I'}\mathbb{E}_{(\boldsymbol{x},y)\sim\mathcal{D}}\left[x_{j'}\right] + \frac{1}{2}$$

And so we get:

$$\left|\mathbb{P}_{(\boldsymbol{z},y)\sim\Gamma_{i\ldots d}(\mathcal{D})}\left[z_j = 1\right] - \frac{1}{2}\right| = \frac{1}{2}\prod_{j'\in I'}\left|\mathbb{E}_{(\boldsymbol{x},y)\sim\mathcal{D}}\left[x_{j'}\right]\right|$$

$$< \frac{1}{2}(1-2\xi)^{|I'|} \le \frac{1}{2} - \xi$$

$\square$

## C.2 AND/OR Circuits

We limit ourselves to circuits where each gate is chosen from the set $\{\wedge, \vee, \neg\wedge, \neg\vee\}$. For every such circuit, we define a generative distribution as follows: we start by sampling a label for the example. Then iteratively, for every gate, we sample uniformly at random a pattern from all the pattern that give the correct output. For example, if the label is $1$ and the topmost gate is OR, we sample a pattern uniformly from $\{(1,1),(1,-1),(-1,1)\}$. The sampled pattern determines what should be the output of the second topmost layer. For every gate in this layer, we sample again a pattern that will result in the correct output. We continue in this fashion until reaching the bottom-most layer, which defines the observed example. Formally, for a given gate $\Gamma \in \{\wedge, \vee, \neg\wedge, \neg\vee\}$, we denote the following sets of patterns:

$$S_\Gamma = \{\boldsymbol{v} \in \{\pm 1\}^2 \ : \ \Gamma(v_1, v_2) = 1\}, \ S_\Gamma^c = \{\pm 1\}^2 \setminus S_\Gamma$$

We recursively define $\mathcal{D}^{(0)}, \ldots, \mathcal{D}^{(d)}$, where $\mathcal{D}^{(i)}$ is a distribution over $\{\pm 1\}^{2^i} \times \{\pm 1\}$:

- $\mathcal{D}^{(0)}$ is a distribution on $\{(1,1),(-1,-1)\}$ s.t. $\mathbb{P}_{\mathcal{D}^{(0)}}\left[(1,1)\right] = \mathbb{P}_{\mathcal{D}^{(0)}}\left[(-1,-1)\right] = \frac{1}{2}$.

- To sample $(\boldsymbol{x},y) \sim \mathcal{D}^{(i)}$, sample $(\boldsymbol{z},y) \sim \mathcal{D}^{(i-1)}$. Then, for all $j \in [2^{i-1}]$, if $z_j = 1$ sample $\boldsymbol{x}'_j \sim U(S_{\gamma_{i,j}})$, and otherwise sample $\boldsymbol{x}'_j \sim U(S_{\gamma_{i,j}}^c)$. Set $\boldsymbol{x} = [\boldsymbol{x}'_1, \ldots, \boldsymbol{x}'_{2^{i-1}}] \in \{\pm 1\}^{2^i}$, and return $(\boldsymbol{x},y)$.

Then we have the following results:

**Lemma 12.** *For every $i \in [d]$ and every $j \in [2^i]$, denote $c_{i,j} = \mathbb{E}_{(\boldsymbol{x},y)\sim\mathcal{D}^{(i)}}\left[x_j y\right]$. Then we have:*

$$|c_{i,j}| - \mathbb{E}\left[y\right] > \left(\frac{2}{3}\right)^d = n^{\log(2/3)}$$

**Lemma 13.** *For every $i \in [d]$ we have $\Gamma_i(\mathcal{D}^{(i)}) = \mathcal{D}^{(i-1)}$.*

Notice that from Lemma 12, the distribution $\mathcal{D}^{(d)}$ satisfies property 1 with $\Delta = n^{\log(2/3)}$ (note that since we restrict the gates to AND/OR/NOT, all gates have influence). By its construction, the distribution also satisfies property 2, and it satisfies property 3 with $\epsilon = \left(\frac{1}{4}\right)^d = \frac{1}{n^2}$. Therefore, we can apply Theorem 2 on the distribution $\mathcal{D}^{(d)}$, and get that algorithm 1 learns the circuit $C$ *exactly* in polynomial time. This leads to the following corollary:

**Corollary 3.** *With the assumptions and notations of Theorem 2, for every circuit $C$ with gates in $\{\wedge, \vee, \neg\wedge, \neg\vee\}$, there exists a distribution $\mathcal{D}$ such that when running algorithm 1 on a sample from $\mathcal{D}$, the algorithm returns $h_C$ with probability $1 - \delta$, in polynomial run-time and sample complexity.*

*Proof.* of Lemma 12 For every $i \in [d]$ and $j \in [2^i]$, denote the following:
$$p_{i,j}^+ = \mathbb{P}_{(\boldsymbol{x},y)\sim\mathcal{D}^{(i)}} [x_j = 1|y = 1], \ p_{i,j}^- = \mathbb{P}_{(\boldsymbol{x},y)\sim\mathcal{D}^{(i)}} [x_j = 1|y = -1]$$
Denote $\mathcal{D}^{(i)}|_{\boldsymbol{z}}$ the distribution $\mathcal{D}^{(i)}$ conditioned on some fixed value $\boldsymbol{z}$ sampled from $\mathcal{D}^{(i-1)}$. We prove by induction on $i$ that $|p_{i,j}^+ - p_{i,j}^-| = \left(\frac{2}{3}\right)^i$:

- For $i = 0$ we have $p_{i,j}^+ = 1$ and $p_{i,j}^- = 0$, so the required holds.

- Assume the claim is true for $i - 1$, and notice that we have for every $\boldsymbol{z} \in \{\pm1\}^{2^{i-1}}$:

$$\mathbb{P}_{(\boldsymbol{x},y)\sim\mathcal{D}^{(i)}} [x_j = 1|y = 1] = \mathbb{P}_{(\boldsymbol{x},y)\sim\mathcal{D}^{(i)}|_{\boldsymbol{z}}} [x_j = 1|z_{\lceil j/2 \rceil} = 1] \cdot \mathbb{P}_{(\boldsymbol{z},y)\sim\mathcal{D}^{(i-1)}} [z_{\lceil j/2 \rceil} = 1|y = 1]$$
$$+ \mathbb{P}_{(\boldsymbol{x},y)\sim\mathcal{D}^{(i)}|_{\boldsymbol{z}}} [x_j = 1|z_{\lceil j/2 \rceil} = -1] \cdot \mathbb{P}_{(\boldsymbol{z},y)\sim\mathcal{D}^{(i-1)}} [z_{\lceil j/2 \rceil} = -1|y = 1]$$

$$= \begin{cases} p_{i-1,\lceil j/2 \rceil}^+ + \frac{1}{3}(1 - p_{i-1,\lceil j/2 \rceil}^+) & if \ \gamma_{i-1,\lceil j/2 \rceil} = \wedge \\ \frac{2}{3}p_{i-1,\lceil j/2 \rceil}^+ & if \ \gamma_{i-1,\lceil j/2 \rceil} = \vee \\ \frac{1}{3}p_{i-1,\lceil j/2 \rceil}^+ + (1 - p_{i-1,\lceil j/2 \rceil}^+) & if \ \gamma_{i-1,\lceil j/2 \rceil} = \neg\wedge \\ \frac{2}{3}(1 - p_{i-1,\lceil j/2 \rceil}^+) & if \ \gamma_{i-1,\lceil j/2 \rceil} = \neg\vee \end{cases}$$

$$= \begin{cases} \frac{2}{3}p_{i-1,\lceil j/2 \rceil}^+ - \frac{1}{3} & if \ \gamma_{i-1,\lceil j/2 \rceil} = \wedge \\ \frac{2}{3}p_{i-1,\lceil j/2 \rceil}^+ & if \ \gamma_{i-1,\lceil j/2 \rceil} = \vee \\ 1 - \frac{2}{3}p_{i-1,\lceil j/2 \rceil}^+ & if \ \gamma_{i-1,\lceil j/2 \rceil} = \neg\wedge \\ \frac{2}{3} - \frac{2}{3}p_{i-1,\lceil j/2 \rceil}^+ & if \ \gamma_{i-1,\lceil j/2 \rceil} = \neg\vee \end{cases}$$

Similarly, we get that:

$$\mathbb{P}_{(\boldsymbol{x},y)\sim\mathcal{D}^{(i)}} [x_j = 1|y = -1] = \begin{cases} \frac{2}{3}p_{i-1,\lceil j/2 \rceil}^- - \frac{1}{3} & if \ \gamma_{i-1,\lceil j/2 \rceil} = \wedge \\ \frac{2}{3}p_{i-1,\lceil j/2 \rceil}^- & if \ \gamma_{i-1,\lceil j/2 \rceil} = \vee \\ 1 - \frac{2}{3}p_{i-1,\lceil j/2 \rceil}^- & if \ \gamma_{i-1,\lceil j/2 \rceil} = \neg\wedge \\ \frac{2}{3} - \frac{2}{3}p_{i-1,\lceil j/2 \rceil}^- & if \ \gamma_{i-1,\lceil j/2 \rceil} = \neg\vee \end{cases}$$

Therefore, we get:

$$|p_{i,j}^+ - p_{i,j}^-| = \frac{2}{3}|p_{i-1,\lceil j/2 \rceil}^+ - p_{i-1,\lceil j/2 \rceil}^-| = \left(\frac{2}{3}\right)^i$$

From this, we get:
$$\left|\mathbb{E}_{(\boldsymbol{x},y)\sim\mathcal{D}^{(i)}} [x_j y]\right| = \left|\mathbb{E}_{(\boldsymbol{x},y)\sim\mathcal{D}^{(i)}} [(2\mathbf{1}\{x_j = 1\} - 1)y]\right|$$
$$= \left|2\mathbb{E}_{(\boldsymbol{x},y)\sim\mathcal{D}^{(i)}} [\mathbf{1}\{x_j = 1\}y] - \mathbb{E}[y]\right|$$
$$= |2 (\mathbb{P}_{\mathcal{D}^{(i)}} [x_j = 1, y = 1] - \mathbb{P}_{\mathcal{D}^{(i)}} [x_j = 1, y = -1])|$$
$$= |2 (p_{i,j}^+ \mathbb{P}[y = 1] - p_{i,j}^- \mathbb{P}[y = -1])|$$
$$= |p_{i,j}^+ - p_{i,j}^-| = \left(\frac{2}{3}\right)^d$$

And hence:

$$\left|\mathbb{E}_{(\boldsymbol{x},y)\sim\mathcal{D}^{(i)}} [x_j y]\right| - \left|\mathbb{E}_{(\boldsymbol{x},y)\sim\mathcal{D}^{(i)}} [y]\right| \geq \left(\frac{2}{3}\right)^d$$

$\square$

*Proof.* of Lemma 13 Fix some $\boldsymbol{z}' \in \{\pm1\}^{n_i/2}$ and $y' \in \{\pm1\}$. Then we have:
$$\mathbb{P}_{(\boldsymbol{x},y)\sim\Gamma_i(\mathcal{D}^{(i)})} [(\boldsymbol{x}, y) = (\boldsymbol{z}', y')] = \mathbb{P}_{(\boldsymbol{x},y)\sim\mathcal{D}^{(i)}} [(\Gamma_i(\boldsymbol{x}), y) = (\boldsymbol{z}', y')]$$
$$= \mathbb{P}_{(\boldsymbol{x},y)\sim\mathcal{D}^{(i)}} [\forall j \ \gamma_{i-1,j}(x_{2j-1}, x_{2j}) = z_j' \ and \ y = y']$$
$$= \mathbb{P}_{(\boldsymbol{z},y)\sim\mathcal{D}^{(i-1)}} [(\boldsymbol{z}, y) = (\boldsymbol{z}', y')]$$

By the definitions of $\mathcal{D}^{(i)}$ and $\mathcal{D}^{(i-1)}$.

$\square$

## Footnotes

[4]We take the sub-gradient zero at zero.