[Reviews · NeurIPS 2020]

Review 1

Summary and Contributions: This paper is about showing how, if distributions exhibit a property called “local correlations”, we may recover the celebrated “weak learning implies strong learning” that holds in the distribution-free setting. This work seeks to add to the discussion surrounding the classical impossibility results by seeking a property of distributions that allows for efficient learning. The contributions include: introducing the “local correlations” concept, illustrated examples of efficient learning with this concept both empirically and theoretically, studying its role in efficient learning in the k-parity problem and in tree-structured boolean circuits, and finally providing a short discussion on characterizing some families of distributions that satisfy the “local correlations” property.

Strengths: The direction of this paper which is influenced by practice is interesting and relevant to NeurIPS. Also, the technical work is sound.

Weaknesses: The paper’s main focus is on learning binary tree-structured circuits. And the “local correlations” property is defined relative to this example. I would have hoped for a more general study or scope of “local correlations”, but then again, perhaps this paper’s main contribution is to just get the conversation started around this property.

Correctness: The developed model seems rigorous. The used methodology is ok.

Clarity: The paper is clearly written.

Relation to Prior Work: The literature is extensively addressed. The approach of the current paper is defended against the previous approached adequately.

Reproducibility: Yes

Additional Feedback: Post Feedback: The authors addressed several of the criticism points in their feedback. The issue of using for the first time the tree-structured Boolean circuits could be further explained. I maintain my review. A more specialized researcher can provide further comments on the feedback response.


Review 2

Summary and Contributions: This paper proposes an distributional assumption called the "Local Correlation Assumption" (LCA), under which weak learning implies efficient strong learning of certain classes of deep functions, including tree-structured Boolean circuits. A layerwise gradient-descent algorithm for learning deep networks that implement tree-structured Boolean circuits is proposed. It is shown that in the (log n)-parity problem, product distributions sufficiently far away from the uniform distribution satisfy the LCA, and thus the proposed algorithm can efficiently learn any target (log n)-parity function. The paper also establishes that for any circuit with AND/OR/NOT gates, there is a generative distribution under which the proposed algorithm will return the exact target function with high probability for a sufficiently large sample.

Strengths: The paper contains a good mix of empirical observations combined with sound theoretical analysis. There is a good effort to explain why the Local Correlation Assumption (LCA) is reasonable, supported with a theoretical analysis of specific families of distributions satisfying the LCA as well as empirical observations of natural data with this property. The technical parts of the main paper appear to be generally correct, and I think the main results are quite insightful. Presentation wise, the main text seems good too, particularly how the LCA was motivated by the example of learning k-parity functions as well as by empirical observations of the ImageNet classification task. Unfortunately, I do not know the broader impact of this work on deep learning research; perhaps this could be judged by experts in deep learning.

Weaknesses: I do not have any major criticism. One question I had was how widely applicable these results are to function classes beyond those of tree-structured circuits and those of circuits with AND/OR/NOT gates (I am not familiar with the deep learning literature). Would it have been interesting to explore the expressive power of functions computed by neural networks with a Boolean circuit structure, as described between lines 131 (page 4) and 147 (page 5); would it be possible to obtain some interesting characterization of such functions? A slight concern I had was the presentation of some proofs in the appendices. While I think the proofs are generally correct (I only went through parts of Appendices A and B), I had trouble parsing some of them due to unexplained and (what seemed to me) conflicting notation (please see "Comments/Suggestions" for details...). There was also notation (such as the definition of f(S) for some f: \chi' \mapsto \chi' and sample S) used before being defined.

Correctness: The main results appear to be generally correct.

Clarity: I think the main text of the paper is generally well-written. Occasionally notation or technical terms in the main text were not defined or defined only in the appendix (please see "Comments/Suggestions" below for details); I suggest looking into this.

Relation to Prior Work: I think there is adequate discussion of the relation between this work and previous related contributions in Section 4. On the subject of Boolean circuits, for example, the current work studies circuits in which the gates may implement any Boolean function, not just the AND/OR/NOT functions (as is often assumed in the literature on Boolean circuits).

Reproducibility: Yes

Additional Feedback: Comments/Suggestions: Page 3, caption for Figure 3: "[Training] depth-two ..." Page 4, line 124: "...contains all the parity [functions] ..." Page 4, equation between lines 136 and 137: Perhaps explain briefly what w and v are (weights?) and what k denotes (size of hidden layer?). What is i? Furthermore, k occurs again in the statements of Theorems 1 and 2 without any explanation (I assumed it denotes the size of the hidden layer, i.e. the number of w_l's). Page 5, Algorithm 1: If we allow updates to V^{(i)} at each time step, how might the bounds obtained in Theorems 1 and 2 be affected? Page 5, line 164: \chi' was defined to be the support of D in Appendix A; I suggest defining \chi' in the main text as well. Page 6, line 207: "...that are not product [distributions]..." Page 6, Theorems 1 and 2: What does it mean that D is "separable by h?" I suggest explaining this term somewhere before stating the theorems. Page 6, property 3: I didn't understand why this property needs to be assumed - could one take \epsilon to be the minimum of P_{(x,y)~D^{(i)}}[(x_{2j-1},x_{2j}) = p] over all i, j and p \in {\pm 1}^2 such that P_{(x,y)~D^{(i)}}[(x_{2j-1}, x_{2j}) = p]? If no such i, j and p exist, then this condition is vacuously satisfied? Was this property stated in order to have a notation for such a lower bound? Page 7, line 224: Where was it assumed earlier that the output of the gate is correlated with the label (i.e. I_{i,j} > 0)? Page 7, line 234: effect -> affect Page 7, Theorem 3: Do we need \xi \leq 1/4? Page 7, line 255: "...from all the [patterns]..." Page 7, line 260: "...such [a] distribution..." Page 8, line 307: "...can take any Boolean [function]..." Page 8, line 313: "...DNFs and [decision] trees..." Page 9, line 327: corrleation -> correlation Organization of Appendices: Should Appendix A (proof of Theorem 2) be subsumed under Appendix B (proofs of Section 3.2)? Lemma 1 precedes Theorem 2 in the main text. Page 12, assignment between lines 439 and 440: By \Psi(S) do you mean the mapping of the instances occurring in S by \Psi (in line 438, \Psi is defined to be a mapping on \chi)? If so, perhaps this could be clarified.. Page 12, line 446: What are \epsilon_d and and \epsilon_{d-1}? Page 12, Lemma 2: I think the notation w_l^{(0)} was defined only later; if so, I suggest moving the definition before the statement of Lemma 2. Also, what does the subscript l index? It does not appear anywhere else in the statement of Lemma 2..? Page 12, Lemma 2: In line 438, the first definition of \Psi(x) suggests that the range of \Psi is contained in [-1,1]^{n_i/2}, but the second definition of \Psi(x) suggests that the range of \Psi is contained in [-1,1]^{n_i} (since the range of \psi is contained in {\pm 1}^{n_i}). Furthermore, \chi' is a subset of {\pm 1}^{2^d}, but in line 440, the expression for B_{W_t^{(i)}, V_0^{(i)}}(x) suggests that \chi' is a subset of {\pm 1}^{2^i}. Page 12, line 446: Here \Psi is assigned the identity map; however, in line 438 \Psi is defined to be some mapping from \chi into [-1,1]^{n_i/2}, which seems to suggest that the dimension of the output should be smaller than that of the input. Page 12, equation between lines 448 and 449: Why is the loss function taken with respect to D here (which is in general unknown) instead of S or \Psi(S) (as in Algorithm 1 and in Lemma 2)? Page 13, third line of equation after line 472: Why do we take ||p|| rather than ||p||^2 here (Shouldn't <(x_{2j-1},x_{2j}), p> = ||p||^2 when (x_{2j-1},x_{2j}) = p)? A similar remark applies to the third line after line 493. Page 14, line 478: Where was it assumed that \tilde{\gamma}(p) = 1? I didn't see this assumption in the statement of Lemma 3... Page 15, line 495: Perhaps explain why \lambda \leq 1. \lambda is defined to be E[y] + \Delta/4; why is the latter expression not more than 1? * I thank the author(s) for the response. Reviewer 4 pointed out some missing literature on read-once formulas; I think it would be good to cite these papers. There were also a few comments on the applicability of tree-structured Boolean circuits to practical networks - perhaps this issue could be further discussed in the paper (space permitting).


Review 3

Summary and Contributions: The paper proposes a property of the data distribution called local correlation assumption for studying provable guarantees of deep learning, which says that small patches of the input and of intermediate layers of the target function are correlated to the target label. This is well justified by a discussion on weak v.s. strong learning and the parity function, with strong empirical evidence from synthetic experiments for the k-parity function and from experiments on real data (e.g., ImageNet). The paper then shows that with the local correlation assumption, when the target function is a tree-structured Boolean circuit (a generalization of k-parity), and the trained network is a deep architecture simulating a tree-structured Boolean circuit with the Boolean gate replaced by a neural gate (a two-layer subnetwork with ReLU + hard tahn gates), there is a gradient-based algorithm that is a provable strong learner.

Strengths: + The topic is interesting. Indeed, most theoretical results on deep learning are not very related to practical scenarios, and the key to bridge the gap is a good model of the property of the practical data. The proposed property in the paper is good progress along the direction. It is natural, can be verified on practical data, and leads to provable guarantees. + The discussion on weak v.s. strong learning and the parity function is insightful.

Weaknesses: - The provable guarantee is given for tree-structured Boolean circuits (and training a network simulating those circuits). These are a quite rich family of functions, but it is unclear if they are that related to practical networks. - The analysis is for layerwise training. Can one show similar guarantees for end-to-end back-propagation? The authors can discuss what the additional challenge for the back-propagation is.

Correctness: I read the analysis, and verify the key lemmas as best as can.

Clarity: Yes

Relation to Prior Work: Yes, clearly discussed.

Reproducibility: Yes

Additional Feedback:


Review 4

Summary and Contributions: The paper proposes an approach to prove positive learnability results for deep learning. In view of the negative results, the challenge in this kind of work is to find assumptions which are natural and tractable. The paper proposes a locality assumption, in the sense that small local patches are correlated with the output. A learnability result is proven for learning Boolean functions computed by a complete binary tree-structured circuit with unknown inner gate functions.

Strengths: The proof of the main learnability result, given in the appendix, is quite involved. I was not able to check the details but the the ideas seem correct and interesting.

Weaknesses: It seems that the assumption of local correlation for the inner gates requires more justification. The kind of locality demonstrated in Figure 1 seems realistic. However, the next steps about a ``target labeling function'' (put in quotation marks in the paper), and then assuming the known tree structure and learning it with a matching neural network, seem less natural and justified. In computational learning theory, learning circuits with having some form of access to the inner structure of the circuit have been considered earlier. Such assumptions can be reasonable in contexts like fault diagnosis, but, as far as I know, are considered somewhat unnatural as a theoretical framework. In this sense the paper seems to start with a realistic assumption, and then prove a technically non-trivial result about an unrealistic function class (with an extended assumption which should be justified in more detail). Is there hope to use the framework without such an assumption?

Correctness: As noted above, the results seem correct (although the details are given in the supplement and I have not verified them in detail).

Clarity: The paper is well written.

Relation to Prior Work: In the related work section it is stated that the authors are ``not aware of any work studying a problem similar to ours''. Tree-structured Boolean circuits are called read-once formulas in the computational learning theory literature (sometimes also called $\mu$-formulas as well), and they form one of the most studied type of Boolean concept classes. Their combinatorial properties are well-studied as well (Golumbic-Gurvich has a survey paper on the topic). While much of the work is in query models (e.g., Angluin-Hellerstein-Karpinski), there are PAC learnability results as well. In particular, Schapire: Learning probabilistic read-once formulas on product distributions (Machine Learning, 1994) studies a probabilistic variant. He also starts with correlations of variables and the target function, and uses partial derivatives. Note that his result does not assume knowing the tree structure, but learns the skeleton as well. I think the relationship with Schapire's work should certainly be clarified. I did not review the whole read-once literature for possible other connections.

Reproducibility: Yes

Additional Feedback: The author response seems rather succinct. In particular, regarding citing the read-once literature, Schapire's paper was suggested to perhaps have a closer connection than just a citation. Also, the examples for known target structure seem to be of a quite different kind.


Review 5

Summary and Contributions: This paper studies the hardness of learning deep neural networks under mild distribution assumption. The main theoretical result is that under teacher-student setting, the hardness of learning certain boolean circuits using deep neural nets can be much better than worst case scenarios under the ''local correlation assumption'', which assumes that outputs of intermediate nodes of the teacher network can be used to weakly learn the labels.

Strengths: 1. This paper supports its ''local correlation assumption'' through an experiment, which shows that the intermediate outputs of a well-trained resnets correlates significantly with the labels. 2. This paper proved that via layer-wise gradient descent, carefully designed neural networks can learn certain boolean circuits using polynomially time and polynomially many samples. This results differs from prior work, which shows in general learning boolean circuits is NP-hard.

Weaknesses: 1. Although the hardness learning boolean circuits is itself an interesting problem, learning boolean circuits can hardly mimic real problems, which limits the significance of the insights for real applications. 2. the results relies on layer-wise training methods, which enables the optimization process but is seldom used in the deep learning community.

Correctness: Basically correct.

Clarity: Overall the paper is clear and well-written. Here are some mild suggestions. 1. Maybe there should be a more cushioned build up before the experiment and technical examples in the first two sections? 2. And maybe the main contributions should be emphasize more in the first section?

Relation to Prior Work: Basically clearly discussed.

Reproducibility: Yes

Additional Feedback: None

[Author Response · NeurIPS 2020]

We thank the reviewers for their overall positive and encouraging feedback.

Responding to the main comments raised by the reviewers:

• Regarding the simplified nature of the target class of tree-structured Boolean circuits: our main goal in the
paper was to suggest a crisp property of "natural" distributions that turns them "easy" to learn. We chose
the class of tree-structured Boolean circuits since this family is rich enough to reveal interesting properties
about learnability using gradient-based algorithms, yet is simple enough to give thorough theoretical analysis.
Showing that the local correlation property plays a role in learning broader families of functions is an exciting
topic for future research.

• On the assumption of known tree structure: we believe that in the context of learning neural networks, knowing
the structure of the tree is a realistic assumptions. Neural networks in practice are often designed to have
a structure that reflects the prior knowledge of the problem (for example, convolutional networks, LSTMs,
transformers etc.).

• Regarding read-once formulas: we will add a comparison to the literature of learning read-once formulas.
However, we note that our focus is not on PAC learnability of read-once formulas in general, but rather on
understanding distributional properties that allow gradient-based algorithms to converge to an optimal solution.

• Regarding the analysis of layerwise training algorithm: we believe that similar results may be shown for
end-to-end optimization, at the expense of making the theoretical analysis much more complicated. End-to-end
gradient-descent is notoriously hard to analyze in the case of deep networks, due to the highly non-convex
nature of the optimization. We chose the less popular layerwise optimization to achieve a simpler theoretical
analysis, and yet show guarantees that reflect the behavior of gradient-descent in practice.

We will correct the final version of the paper according to additional comments and suggestions raised by the reviewers.

[Meta-Review · NeurIPS 2020]

The reviews all agree that the paper is well-written, the paper explains clearly that LCA is a reasonable assumption both empirically and theoretically. Moreover, the paper shows a novel result indicating weak learners for distributions with LCA property can be transferred into strong learners. This could have potential impact in deep learning as well.